# SEAFormer: A Spatial Proximity and Edge-Aware Transformer for Real-World Vehicle Routing Problems

## Abstract

Real-world Vehicle Routing Problems (RWVRPs) require solving complex, sequence-dependent challenges at scale with constraints such as delivery time window, replenishment or recharging stops, asymmetric travel cost, etc. While recent neural methods achieve strong results on large-scale classical VRP benchmarks, they struggle to address RWVRPs because their strategies overlook sequence dependencies and underutilize edge-level information, which are precisely the characteristics that define the complexity of RWVRPs. We present SEAFormer, a novel transformer that incorporates both node-level and edge-level information in decision-making through two key innovations. First, our Clustered Proximity Attention (CPA) exploits locality-aware clustering to reduce the complexity of attention from $O(n^2)$ to $O(n)$ while preserving global perspective, allowing SEAFormer to efficiently train on large instances. Second, our lightweight edge-aware module captures pairwise features through residual fusion, enabling effective incorporation of edge-based information and faster convergence. Extensive experiments across four RWVRP variants with various scales demonstrate that SEAFormer achieves superior results over state-of-the-art methods. Notably, SEAFormer is the first neural method to solve 1,000+ node RWVRPs effectively, while also achieving superior performance on classic VRPs, making it a versatile solution for both research benchmarks and real-world applications.

## 1 Introduction

The Vehicle Routing Problem (VRP) is a fundamental challenge in logistics, where the goal is to deliver goods to many customers from a depot while minimizing cost and respecting constraints such as vehicle capacity. In practice, VRPs appear most prominently in last-mile deliveries, which have surged dramatically in recent years. For example, in 2022, Manhattan saw over 2.4 million delivery requests per day (Blueprint, 2022), averaging over 3,000 deliveries per minute during a 12-hour workday. Although existing solutions perform well on simplified benchmarks, they rarely account for the operational real-world problems, scales, and constraints encountered in practice.

Real-world Vehicle Routing Problems (RWVRPs) extend the VRP and includes variants such as VRPTW (time window (Kallehauge et al., 2005)), EVRPCS (electric vehicles with recharging - which is important as carrying large weight significantly reduces their driving range (Szumska et al., 2021)), VRPRS (replenishment stops where vehicles can restock to continue service (Schneider et al., 2015)) and AVRP (asymmetric travel costs (Vigo, 1996)). RWVRPS incorporate sequence-dependent constraints and have the following properties: **i) Local infeasibility:** A decision's validity cannot be determined in isolation, and it depends on the complete path history. **ii) State accumulation:** Vehicle state (battery level, current time, etc.) evolves dynamically through the route and is only calculable when the visitation sequence is defined. **iii) Tightly coupled constraints:** Violating one constraint propagates through subsequent decisions. These interdependencies transform the capacity-constrained spatial optimization problem of simple VRPs into a sequence-dependent one and form a tightly coupled setup that can only be validated by considering the entire route sequence.

Recent progress in deep learning has delivered impressive results for combinatorial optimization, reaching near-optimal solutions on classical VRP benchmarks. Existing approaches can be grouped

into two main paradigms: autoregressive and non-autoregressive models, each offering unique benefits and facing specific challenges when extended to RWVRPs.

Autoregressive methods (Kool et al., 2018; Kwon et al., 2020; Berto et al., 2025; Li et al., 2024; Lin et al., 2021; Chen et al., 2022; Wang et al., 2024) designed for VRP or RWVRPs, construct solutions sequentially using transformers and perform well on small-scale benchmarks. However, their performance deteriorates as problem size increases. They lack spatial inductive bias, treating all node pairs uniformly, disregarding the inherent geometric structure of routing problems. In addition, their full attention incurs an $O(n^2d)$ memory cost, where $d$ is the embedding dimension, limiting training to hundreds rather than thousands of nodes. Moreover, they cannot represent edge-specific attributes (e.g., asymmetric costs, energy use, or traffic), limiting applicability to real-world routing where pairwise relationships drive feasibility and optimality. These architectural limitations make current autoregressive methods unsuitable for deployment on industrial-scale real-world routing problems.

Recent efforts to address the scalability limitations of autoregressive models introduce new challenges when applied to RWVRPs. Divide-and-conquer methods (Hou et al., 2022; Nasehi et al., 2025; Zheng et al., 2024) decompose the problem by clustering customers first and then solving each cluster independently. However, in RWVRPs this creates a fundamental circular dependency: assessing whether a cluster is feasible requires knowing the vehicle's state upon arrival (e.g., remaining battery in EVRPCS or current time in VRPTW), but that state depends on the full route, which is not yet known during clustering. This mismatch often leads to infeasible clusters that require costly post-hoc repair or prevents the discovery of high-quality routes altogether (refer to Appendix G). Other scalable architectures (Gao et al., 2024; Luo et al., 2023; 2025a) face their own limitations, including substantial computational overhead or reliance on supervised learning which requires a set of pre-computed optimal solutions that are themselves computationally expensive to obtain.

In contrast, non-autoregressive methods (Kool et al., 2022; Qiu et al., 2022) improve scalability by predicting all routing decisions simultaneously through learned heatmaps, thereby avoiding the sequential bottleneck of autoregressive decoding. While computationally efficient, these methods face a fundamental challenge similar to divide-and-conquer approaches: they cannot assess constraint satisfaction without the knowledge of the traversal sequence. Critical constraints, such as battery levels, are inherently path-dependent and require sequential state accumulation to ensure feasibility. As a result, non-autoregressive models are inherently hard to adapt to RWVRPs.

In this paper, we introduce **SEAFormer** (Spatial proximity and Edge-Aware transFormer), a novel architecture that combines the representational power required for RWVRPs with the computational efficiency necessary for real-world use. Unlike existing methods that treat routing as purely node selection, our model explicitly reasons about both *where to go* (nodes) and *how feasible/costly that transition is* (edges), a distinction critical for handling edge-level information and improving generalization. SEAFormer introduces two complementary innovations that together enable scalable, high-quality solutions across diverse RWVRP and VRP variants.

First, we propose **Clustered Proximity Attention (CPA)**, a spatially aware attention mechanism that improves generalization while reducing the complexity of full attention from $O(n^2)$ to $O(n)$. Unlike generic sparse attention methods (e.g., Longformer (Beltagy et al., 2020)), CPA leverages problem-specific spatial patterns to cluster nodes into meaningful partitions and applies attention within each partition. Our deterministic-yet-diverse clustering strategy offers multiple spatial perspectives, to avoid the local optima issues often encountered in sparse attention mechanisms.

Second, we introduce a **lightweight edge-aware module** that explicitly models pairwise relationships between nodes, capabilities largely absent from existing approaches. While node embeddings capture individual locations and demands, they cannot represent edge-specific attributes. Incorporating edge-level information not only enables solutions to problems where such details are essential but also enhances accuracy, generalization, and convergence across models. Our module learns these relational patterns through a parameter-efficient architecture (increasing number of parameters by only 7.5%), which is additively combined with the attention decoder to jointly optimize the problem. SEAFormer is the first learning-based approach to effectively solve 1000+ node instances across RWVRPs within a single architecture, achieving competitive or superior performance consistently.

We evaluate SEAFormer on four RWVRP variants across a wide range of problem sizes and compare it with state-of-the-art neural and classical methods. For complex variants such as VRPTW, EVRPCS, VRPRS, and AVRP, it delivers at least 15% reduction in the objective value over the

large-scale instances while fully respecting operational constraints. On standard VRP benchmarks, SEAFormer consistently surpasses the state-of-the-art approaches. Even on large-scale instances, SEAFormer preserves solution quality, whereas existing neural solvers often run out of memory or suffer substantial performance degradation.

The contributions of this paper are threefold. (i) We introduce SEAFormer, the first transformer to jointly optimize spatial proximity and edge-level constraints for real-world VRP variants, achieving state-of-the-art performance across different benchmarks. (ii) We propose Clustered Proximity Attention (CPA), a problem-specific sparse attention mechanism that leverages spatial locality in routing tasks, enhancing generalization and reducing memory complexity through deterministic-yet-diverse clustering. (iii) We propose a parameter-efficient edge-aware module that integrates pairwise relational information into routing decisions, enabling seamless handling of edge-specific constraints that are essential for real-world deployment yet missing from existing neural solvers.

## 2 RELATED WORK

We review existing learning-based approaches to RWVRPs and VRP, focusing on why they struggle to simultaneously achieve solution quality, computational efficiency, and constraint satisfaction at real-world scales. A more comprehensive discussion is provided in Appendix J.

### 2.1 NEURAL SOLVERS WITH LIMITED SCALABILITY

Autoregressive neural methods construct solutions sequentially using learned policies, with several notable approaches in the literature (Kool et al., 2018; Kwon et al., 2020; Berto et al., 2025; Luo et al., 2023). These models capture sequence dependencies during solution generation. While effective on small-scale problem instances, their performance deteriorates on large-scale problems. Moreover, reliance on the full-attention mechanism limits scalability and prevents efficient iterative refinement on standard hardware.

### 2.2 SCALABLE NEURAL APPROACHES

Recent approaches for scalable VRP solutions (Hou et al., 2022; Zheng et al., 2024; Nasehi et al., 2025; Ye et al., 2024) are based on divide-and-conquer method. Such techniques, however, cannot solve RWVRPs effectively as they cluster nodes before determining visit sequences, yet feasibility of a solution for RWVRPs depends on those sequences. Whether a vehicle can serve a cluster requires knowing its state upon arrival, such as battery level in EVRPCS or elapsed time in VRPTW, which is unavailable during clustering. This circular dependency causes clusters to violate constraints once sequenced, requiring expensive repairs or prevents finding high-quality solutions.

Scalable VRP solutions that do not rely on divide-and-conquer strategy face challenges when applied to RWVRPs. ELG (Gao et al., 2024) restricts attention to nearby nodes using distance-based penalties, which can limit performance in RWVRPs where vehicles need to reach distant nodes. Furthermore, the local attention approach does not explicitly account for RWVRP-specific constraints, which may lead to infeasible or suboptimal solutions. Heavy decoder architectures (Luo et al., 2023; 2025a) rely on supervised learning from near-optimal solutions. For RWVRPs, generating such training data is computationally expensive. For example, solving 100-node EVRPCS instances to optimality can be prohibitively time-consuming, and in some cases computationally infeasible, making these approaches impractical for RWVRPs.

Non-autoregressive models Ye et al. (2024); Qiu et al. (2022); Kool et al. (2022) generate high-quality solutions for large-scale VRPs by predicting edge inclusion probabilities via a heatmap. However, they face specific challenges with sequential constraints in RWVRPs. For instance, in EVRPCS, edge feasibility depends on the vehicle's battery state upon arrival, information that is unavailable during parallel prediction. Applying non-autoregressive methods to sequence-dependent problems requires costly repair procedures too, which often leads to a drop in solution quality.

#### 2.2.1 RWVRP-SPECIFIC METHODS

A few recent works directly address variations of RWVRPs. Lin et al. (2021) used Transformer-LSTM to track electric vehicle travel history, while Chen et al. (2022) employed GRUs with two-

stage training to improve charging station routing. Wang et al. (2024) introduced a GAT-based encoder with penalty functions to enforce constraints for EVRPCS. Liu et al. (2024) and Berto et al. (2025) proposed multi-task learning for VRP variants such as VRPTW. Despite these advances, existing methods face at least one of the following limitations: (i) limited scalability, resulting in low-quality solutions for instances larger than 100 nodes; (ii) reliance on manually tuned penalty terms, which slows training and may still produce infeasible solutions requiring costly post-processing; and (iii) sub-optimal strategies for visiting infrastructure nodes, limiting the model's ability to determine when and where to charge or restock efficiently, leading to lower quality solutions.

## 2.3 SPARSE ATTENTION MECHANISMS

Recent advances in efficient attention offer potential solutions to scalability challenges. Reformer (Kitaev et al., 2020) leverages locality-sensitive hashing, Longformer (Beltagy et al., 2020) employs sliding windows, and BigBird (Zaheer et al., 2020) combines random, window, and global attention. These mechanisms are designed for text processing and do not naturally align with the geometric structure of routing problems and thus cannot be easily extended to spatial problems. While language tasks mainly involve local sequential dependencies, routing requires radial connectivity between depots and customers. Sparsity patterns based on these methods will disrupt the spatial relationships, limiting performance on routing tasks.

## 3 CLUSTERED PROXIMITY ATTENTION

Figure 1 shows a VRP solution captured from Hou et al. (2022), highlighting an important structural pattern: nodes within the same route tend to cluster based on their polar coordinates relative to the depot. In particular, they typically follow one of three patterns: (1) similar angles but varying distances from the depot, (2) similar distances but different angles, or (3) close proximity in both angle and distance. This insight motivates our polar clustering approach, as optimal routes naturally reflect these geometric relationships.

We propose **Clustered Proximity Attention (CPA)**, a problem-specific sparse attention mechanism that maintains spatial routing structure while substantially reducing computational complexity. The procedure is detailed below.

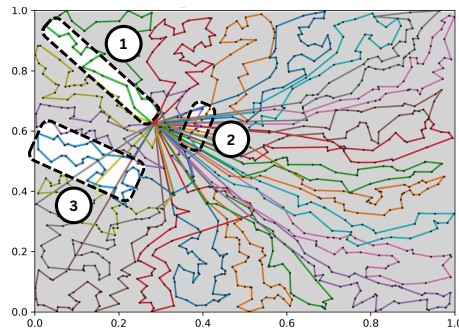

Figure 1: A VRP solution. Nodes within the same route can exhibit: (1) similar angles but different distances, (2) similar distances but different angles, or (3) a close proximity in both.

**Polar-based Spatial Transformation.** Given node coordinates $\{x_i \in \mathbb{R}^2\}_{i=0}^n$ with depot $n_0$, first we transform each customer location to polar coordinates:

$$r_i = \|x_i - x_0\|_2, \quad \theta_i = \arctan 2(y_i - y_0, x_i - x_0) \in [0, 2\pi), \tag{1}$$

where $r_i$ represents radial distance and $\theta_i$ represents angular position relative to the depot.

**Partitioning Score.** To form clusters capturing the diverse spatial patterns illustrated in Figure 1 and necessary for optimal routing, we introduce a partitioning score that balances radial and angular proximity. After normalizing polar coordinates to $[0, 1]$, we compute a clustering metric as:

$$s_i^{(\alpha)} = \alpha \cdot \bar{\theta}_i + (1 - \alpha) \cdot \bar{r}_i, \tag{2}$$

where $\bar{r}_i = (r_i - r_{\min})/(r_{\max} - r_{\min})$ and $\bar{\theta}_i = \theta_i/2\pi$ represent the normalized radial and angular coordinates, respectively, and $\alpha$ is the mixing coefficient that controls the relative importance of distance versus angle in cluster formation.

**Deterministic-yet-Diverse Partitioning.** To prevent overfitting to specific cluster configurations, preserve global context, and capture diverse spatial patterns while maintaining training efficiency, we use $R$ partitioning rounds with varied mixing coefficients, defining $\alpha$ in Equation 2 as:

$$\alpha = t/(R-1), \quad t \in \{0, 1, \dots, R-1\}. \tag{3}$$

This creates a spectrum from radial ($\alpha = 0$, grouping nodes at similar distances from depot) to angular clustering ($\alpha = 1$, grouping nodes sharing a similar angle from the depot). Each configuration captures different spatial patterns shown in Figure 1, entails to a proper global prospective.

**Boundary Smoothing for Robust Clustering.** Hard cluster boundaries can artificially separate nearby nodes, disrupting natural customer groups. We address this through a boundary smoothing technique. After sorting the calculated partitioning scores $S_\alpha = [s_\alpha^1, \dots, s_\alpha^n]$, given $M$ as the cluster size (a user-defined parameter analyzed in Appendix F), we apply a circular shift as:

$$S'_\alpha = [s_\alpha^{\lfloor M/2 \rfloor + 1}, \dots, s_\alpha^n, s_\alpha^1, \dots, s_\alpha^{\lfloor M/2 \rfloor}] \tag{4}$$

This rotation softens the deterministic cluster boundaries that would otherwise push nearby nodes apart, keeping close nodes together and improving transitions between clusters. After computing the partitioning scores $S = \{S_{\alpha_1}, S'_{\alpha_1}, S_{\alpha_2}, \dots\}$ across different proximity weights and rotations, we partition nodes into clusters of size $M$ using the partitioning score. Note that when $n$ is not perfectly divisible by (i.e., when $\lceil n/M \rceil \times M > n$, we pad the final cluster with depot nodes to ensure uniform cluster size $M$ across all partitions).

**Localized Attention Computation.** Given partition $\mathcal{P} = \{C_{\alpha_1}^1, C_{\alpha_1}^2, \dots, C_{\alpha_1}^k, \dots\}$ where each $C_\alpha^j$ contains $M$ proximate nodes, we compute attention independently within each cluster as:

$$\text{Attention}(Q_j, K_j, V_j) = \text{softmax}\left(\frac{Q_j K_j^\top}{\sqrt{d}}\right) V_j, \quad \text{where } Q_j, K_j, V_j \in \mathbb{R}^{|C_\alpha^j| \times d}. \tag{5}$$

**Complexity analysis:** CPA reduces complexity of attention from $O(n^2)$ to $O(n)$ while preserving the spatial relationships critical for routing decisions. It partitions nodes into clusters of size $M$. Each cluster requires $O(M^2)$ operations for attention computation. With $\lceil n/M \rceil$ clusters total:

$$\text{CPA complexity} = O\big(\underbrace{R}_{\text{No. rounds}} \times \underbrace{\lceil n/M \rceil}_{\text{clusters}} \times \underbrace{(M^2)}_{\text{per cluster}}\big) = O(nRM) = O(n) \tag{6}$$

### 3.1 VISUAL EXAMPLE OF CPA

Figure 2 illustrates the steps of CPA. Figure 2a shows the problem instance, where customer nodes (blue dots) are distributed in a 2D Euclidean space around the depot (red star). The instance is then transformed into polar coordinates relative to the depot (Figure 2b). Next, the partitioning score from Equation 2 is computed (Figure 2c), after which the nodes are sorted by this score and grouped into clusters based on the selected cluster size (Figure 2d). Finally, Figure 2e compares CPA's attention scores with standard full attention (Figure 2f), highlighting CPA's locality-aware sparsity and reduced memory footprint.

## 4 SEAFORMER ARCHITECTURE

Figure 3 shows the SEAFormer architecture, a dual-stream design that jointly models node-level spatial structure and edge-level information for RWVRPs. The node stream encodes location features using our Clustered Proximity Attention (presented in Section 3), while the edge-aware stream maintains explicit embeddings for pairwise attributes such as costs and distances. These two streams converge in the decoder: node embeddings drive the sequential step-by-step node selection, and edge embedding is used to rank candidate transitions from the current position through a learned heatmap generation. This separation of roles allows SEAFormer to scale to thousands of nodes, converge faster, and remain expressive enough to handle complex real-world constraints. The encoder of SEAFormer comprises two complementary modules.

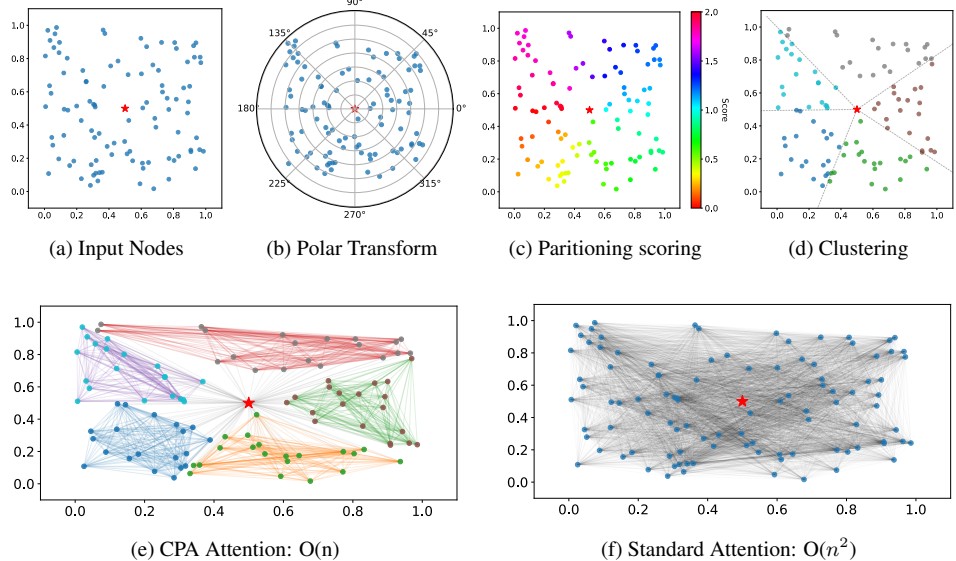

Figure 2: Pipeline of CPA. (a) Input nodes in Cartesian space with depot (red star), (b) Polar transformation relative to depot, (c) Angular scoring with $\alpha = 0$ (pure angular), (d) Nodes sorted by angle and partitioned into fixed-size clusters, (e) Final CPA attention pattern with O(n) complexity, (f) Standard attention with O($n^2$) complexity showing all pairwise connections. CPA reduces number of attention calculation for this example from 10,000 to 2100 (approximately 80% reduction).

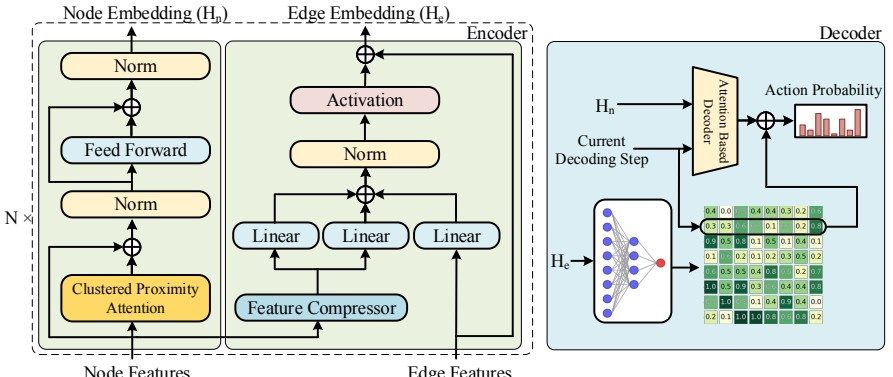

Figure 3: SEAFormer Architecture. The dual-module encoder embeds nodes through CPA and edges through a lightweight residual module. The dual-path decoder combines edge-aware guidance (heatmap) with sequential node selection (attention), unified through logit fusion. This design enables scalable training on large number of nodes while handling diverse RWVRP constraints.

**Node embedding through CPA.** This module adopts the encoder architecture from Kwon et al. (2020), encoding node features through $L$ layers. Unlike prior works that employ full attention mechanisms, SEAFormer utilizes CPA to produce spatially-aware node embeddings $H_n \in \mathbb{R}^{n \times d}$, while achieving significant memory savings in training and inference.

For problem variants with optional service nodes such as EVRPCS (charging stations) and VRPRS (replenishment stops), we augment the spatial encoder with a specialized optional node processing pathway. This parallel encoding layer generates embeddings for optional nodes $H_o \in \mathbb{R}^{f \times d}$, where

$f$ represents the number of service facilities. These embeddings are then fused with customer embeddings through a learnable mechanism, and the combined representations pass through the same batch normalization and feed-forward layers. This allows each customer embedding to account for nearby optional facilities and their influence on route feasibility, which is crucial in problems where strategic service stops can substantially enhance or even enable high-quality solutions. A comprehensive description of the optional node embedding is provided in Appendix H.

**Edge-Aware Embedding Module.** We design our edge embedding to preserve the properties that exist for edges while maintaining computational efficiency. The edge-aware embedding module only operates on edges between depots and customers, while optional nodes in EVRPCS and VRPRS do not participate in this process. Given edge features (distance, energy consumption rate, historical traffic) embedding $X_e^{(i,j)} \in \mathbb{R}^{d_e}$, and node embeddings $H_n^{(i)}, H_n^{(j)} \in \mathbb{R}^d$, we compute edge-aware embeddings through residual fusion (Szegedy et al., 2017):

$$H_e^{(i,j)} = X_e^{(i,j)} + \sigma \left( \text{BN} \left( W_1 H_n^{(i)} + W_2 H_n^{(j)} + W_3 X_e^{(i,j)} \right) \right) \tag{7}$$

Where $W$ is a trainable parameter, $W_1 H_n^{(i)}$ captures origin-specific factors, $W_2 H_n^{(j)}$ captures destination-specific factors, $W_3 X_e^{(i,j)}$ captures edge features. Prior to this, we apply a linear transformation to reduce node features to match edge embedding dimensions, enabling the model to extract only relative information from node embeddings. Batch normalization ensures stable training across diverse edge scales and SILU activation ($\sigma$) enables modeling of complex non-linear relationships. This architecture anchors edge representations in spatial context while increasing the model's total parameters by only 7.5%. Lastly, a residual connection maintains original edge information while selectively integrating node-level context.

Once edge and node embeddings are generated, SEAFormer's decoder constructs the solution through dual complementary pathways that balance global optimization with sequential decoding.

**Edge-Guided Global Heatmap.** Before sequential decoding, we produce a static heatmap that encodes global edge preferences. This step occurs once at the onset of the decoding process:

$$\mathcal{H}_{ij} = tanh\big(\text{MLP}_\theta(H_e^{(i,j)})\big) \in \mathbb{R}, \quad \forall i, j \in \{1, \ldots, n\}, \tag{8}$$

where $\text{MLP}_\theta$ is a multi-layer perceptron that maps each edge to a scalar score, followed by the $tanh$ activation function. This module encodes edge features (e.g., travel times, distances) that are impossible to capture through node-level encoding.

**Node-Guided Sequential Attention.** The sequential decoder constructs solutions autoregressively, selecting one node at each step $t$ based on the current state of the partially generated solution. To enable the model to differentiate between vehicle states across problem variants, we adapt the query vector to each RWVRP variant's state representation $\mathbf{q}_t = [\mathbf{h}_t; c_t; \xi_t]$, where $\mathbf{h}_t$ is the last visited node embedding and $\xi_t$ represents variant-specific constraints: none for VRP, battery level $b_t$ for EVRPCS, remaining travel length $\tau_t$ for VRPRS, and current time $w_t$ for VRPTW. To ensure stable convergence and prevent bias from large values, we normalize $\xi_t$ by its maximum value to maintain the range $[0, 1]$. Finally, the attention mechanism computes compatibility scores with all unvisited nodes through:

$$\alpha_{ti} = \begin{cases} \frac{\exp(\mathbf{q}_t^\top \mathbf{k}_i / \sqrt{d})}{\sum_{j \in \mathcal{U}_t} \exp(\mathbf{q}_t^\top \mathbf{k}_j / \sqrt{d})} & \text{if } i \in \mathcal{U}_t \\ 0 & \text{otherwise} \end{cases} \tag{9}$$

where $\mathcal{U}_t$ denotes the set of unvisited feasible nodes. We implement a proactive masking function that enhances solution quality, reduces search space, and prevents the model from generating infeasible solutions during inference (see Appendix B for detailed description). Finally, at each step $t$, we combine logits from both paths:

$$\ell_t = \ell_t^{\text{seq}} + \mathcal{H}_{i_t,:}, \tag{10}$$

where $\ell_t^{\text{seq}}$ is sequential attention logits, $\mathcal{H}_{i_t,:}$ is the heatmap row for current node $i_t$. For optional nodes not included in the heatmap, we define heatmap values as $\mathcal{H}_{i_t,j} = -2 \cdot \frac{r_t}{R_{\max}}, \quad j \in \mathcal{O}$ to encourage model to visit such nodes more frequently as resources deplete. Here, $r_t$ denotes the remaining resource level (battery charge in EVRPCS or available driving range in VRPRS), $R_{\max}$ represents the maximum resource capacity (full battery in EVRPCS or max driving range in VRPRS), and $\mathcal{O}$ is set of optional nodes.

# 5 EXPERIMENTS

To verify the applicability of SEAFormer on different variations of RWVRP and VRP, we evaluate SEAFormer on 5 combinatorial optimization problems, including VRPTW, EVRPCS, VRPRS, AVRP, and VRP. Detailed formulations of these problems are provided in Appendix A.

**Benchmarks.** We evaluate SEAFormer across four datasets: (i) Random RWVRP benchmarks, comprising 100 instances per problem size with up to 7K nodes (results up to 1k are shown in table 1, and larger values in appendix C) following Kwon et al. (2020); Zhou et al. (2023); (ii) Random VRP benchmarks with up to 7K nodes following Zhou et al. (2023); (iii) real-world VRP instances from CVRPLib; (iv) cross-distribution datasets generated by Zhou et al. (2023).

**Implementation.** For all problems, we train the model for 2000 epochs on 100-customer instances, then 200 epochs on 500-customer instances, and 100 epochs on 1000-customer instances. As in prior work, problem instances are uniformly sampled from the $[0, 1]^2$ space, and demands drawn from a discrete uniform distribution on $[1, 10]$. Asymmetric VRPs are created using our dataset generation procedure (refer to appendix A.4.1). The Adam optimizer (Kingma, 2014) is employed for training, with detailed hyperparameters for each problem instance as well as training times provided in Appendix K and Appendix L.

**Metrics.** We report the mean objective (Obj.), gap (G), and inference time (T) for each approach. Objective represents solution length, where lower values signify superior performance. Gap quantifies the deviation from solutions generated by one of OR-tools, LKH, or HGS. Time denotes the total runtime across the entire dataset, measured in seconds (s), minutes (m), or hours (h). Runtimes, measured on identical hardware (NVIDIA A100 GPU for neural methods, 32-core CPU for classical solvers), exclude model loading and represent the total solution time over each dataset.

**Inference.** In the inference phase, we evaluate SEAFormer using two strategies. First, greedy decoding with 8-fold augmentation, and second, Simulation-Guided Beam Search (SGBS) (Choo et al., 2022), which requires additional computation time but yields superior results as it explores multiple solution trajectories simultaneously. The greedy variant of SEAFormer operates significantly faster than the SGBS version at the expense of marginally reduced solution quality.

**Baselines.** We compare SEAFormer with **1) Classical Solvers:** HGS-PyVRP (Wouda et al., 2024), LKH (Helsgaun, 2017), and OR-Tools; **2) Construction-based NCO Methods:** POMO (Kwon et al., 2020), MTPOMO (Liu et al., 2024), EVRPRL (Lin et al., 2021), EVGAT (Wang et al., 2024), LEHD (Luo et al., 2023), RELD (Huang et al., 2025), ELG (Gao et al., 2024) L2C-insert (Luo et al., 2025b), UniteFormer (Meng et al.), Eformer (Meng et al., 2025) DAR (Wang et al., 2025), BLEHD (Luo et al., 2025a), and RouteFinder Berto et al. (2025); **3) Divide-and-conquer based methods:** GLOP (Ye et al., 2024), DeepMDV (Nasehi et al., 2025), and UDC (Zheng et al., 2024). Not all existing methods apply to every RWVRP variant, so comparisons are limited to applicable approaches. For details on baselines and their integration, see Appendix M.

## 5.1 EVALUATION ON RWVRPS

Table 1 presents a comprehensive evaluation of SEAFormer against state-of-the-art methods across four challenging RWVRPs. SEAFormer consistently achieves superior or competitive performance across all problem variants and scales. Notably, using SGBS as a search method, SEAFormer establishes new state-of-the-art results in learning-based methods across all 12 test configurations between, at the cost of higher processing time, particularly with impressive gains on larger instances.

While SEAFormer achieves superior performance on VRPTW, EVRPCS, and VRPRS across all settings where existing large-scale solutions fail, the AVRP results especially showcase SEAFormer's architectural strengths. Whereas learning-based approaches including POMO, LEHD, BLEHD, and UDC struggle with asymmetric distance matrices and show gaps up to 3% with OR-Tools on 100-customer instances, SEAFormer achieves the best performance with a 1.7% improvement over OR-Tools and increasingly larger gains on bigger instances. This highlights our model's superior capacity to capture complex spatial dependencies inherent in asymmetric routing problems.

The results highlight SEAFormer's exceptional scalability: while most baselines suffer severe degradation on 1,000-customer RWVRPs, SEAFormer preserves solution quality and remains the fastest method. The SGBS variant, though requiring additional computation (95-100 minutes for 1k cus-

Table 1: Objective function (Obj.), Gap to the OR-tools (Gap), and solving time (Time) on 100, 500, and 1,000-node RWVRPs. All test sets contain 100 instances following settings in Zhou et al. (2023). The overall best performance is in bold and the best learning method is marked by shade. OR-Tools results are not optimal, as execution was stopped early due to time limits. Methods not tailored to EVRPCS or VRPRS are excluded due to sub-optimal performance. (Refer to Section G).

| RWVRP | METHODS | 100 CUSTOMERS | | | 500 CUSTOMERS | | | 1K CUSTOMERS | | |
|---|---|---|---|---|---|---|---|---|---|---|
| | | OBJ.↓ | G(%) | T | OBJ.↓ | G(%) | T | OBJ.↓ | G(%) | T |
| VRPTW | OR-TOOLS | 26.34 | 0.00 | 1H | 87.3 | 0.00 | 5H | 151.4 | 0.00 | 10H |
| | HGS-PYVRP | **26.04** | **-1.1** | 1H | **83.8** | **-4.00** | 5H | **142.6** | **-5.6** | 10H |
| | POMO | 26.81 | 1.78 | 5S | 93.2 | 6.75 | 30S | 193.2 | 27.6 | 1M |
| | DAR | 27.1 | 2.88 | 5S | 97.2 | 11.3 | 30S | 212.2 | 40.1 | 1M |
| | MTPOMO | 27.02 | 2.58 | 5S | 96.8 | 10.8 | 30S | 207.1 | 36.7 | 1M |
| | ROUTEFINDER | 26.8 | 1.74 | 5S | 91.2 | 4.46 | 30S | 171.4 | 11.3 | 1M |
| | SEAFORMER | 26.75 | 1.55 | 5S | 87.4 | 0.11 | 30S | 149.7 | -1.1 | 1M |
| | SEAFORMER-SGBS | 26.5 | 0.6 | 10S | 85.0 | -2.6 | 13M | 145.2 | -4.1 | 1.6H |
| EVRPCS | OR-TOOLS | 16.35 | 0.00 | 1H | 31.1 | 0.00 | 5H | 47.8 | 0.00 | 10H |
| | EVRPRL | 16.54 | 1.16 | 10S | 34.3 | 10.3 | 90S | 62.1 | 29.9 | 3M |
| | EVGAT | 16.9 | 3.36 | 30S | 35.2 | 13.2 | 3M | 56.3 | 17.8 | 6M |
| | SEAFORMER | 16.36 | 0.06 | 5S | 30.8 | -1.0 | 30S | 45.8 | -4.2 | 1M |
| | SEAFORMER-SGBS | **16.14** | **-1.3** | 15S | **30.2** | **-2.9** | 13M | **44.9** | **-6.1** | 1.7H |
| VRPRS | OR-TOOLS | 11.2 | 0.00 | 1H | 23.4 | 0.00 | 5H | 36.11 | 0.00 | 10H |
| | EVRPRL | 11.43 | 2.05 | 10S | 25.16 | 7.52 | 90S | 47.5 | 31.5 | 3M |
| | EVGAT | 11.76 | 5.0 | 30S | 25.65 | 9.6 | 3M | 42.1 | 16.6 | 6M |
| | SEAFORMER | 11.24 | 0.35 | 5S | 22.93 | -2.0 | 30S | 34.87 | -3.4 | 1M |
| | SEAFORMER-SGBS | **10.97** | **-2.0** | 15S | **22.33** | **-4.6** | 13M | **33.95** | **-6.0** | 1.7H |
| AVRP | OR-TOOLS | 19.37 | 0.00 | 1H | 40.27 | 0.00 | 5H | 47.6 | 0.00 | 10H |
| | POMO | 19.5 | 0.7 | 10S | 42.12 | 4.59 | 20S | 53.44 | 12.3 | 1M |
| | MTPOMO | 19.63 | 1.3 | 10S | 44.4 | 10.2 | 3M | 56.2 | 18.1 | 1M |
| | ROUTEFINDER | 19.6 | 1.2 | 5S | 41.52 | 3.1 | 20S | 48.3 | 1.47 | 1M |
| | LEHD | 19.96 | 3.0 | 5S | 40.71 | 1.1 | 20S | 45.89 | -3.6 | 1.2M |
| | BLEHD-PRC50 | - | - | - | 42.13 | 4.61 | 2.5M | 45.23 | -5.0 | 6M |
| | UDC$_{250}(\alpha = 50)$ | - | - | - | 40.06 | -0.5 | 30M | 45.17 | -5.1 | 1.2H |
| | SEAFORMER | 19.48 | 0.6 | 5S | 40.23 | -0.1 | 30S | 45.14 | -5.2 | 1M |
| | SEAFORMER-SGBS | **19.04** | **-1.7** | 10S | **38.97** | **-3.2** | 13M | **44.07** | **-7.4** | 1.7H |

tomers), consistently produces the best solutions across all scales, suggesting an effective trade-off between solution quality and computational resources.

## 5.2 EVALUATION ON VRP

Table 2 reports SEAFormer's performance on VRP instances with 100–1000 customers, showing both strong scalability and high solution quality. On small instances, SEAFormer achieves 0.96% gap in 5 seconds, and with SGBS it matches POMO, and slightly outperform UniteFormer and Eformer. The advantage becomes clearer at larger scales: on 500-customer instances, SEAFormer achieves a 4.64% gap in 15 seconds, which outperforms POMO and rivaling methods like GLOP-LKH3. With SGBS, SEAFormer further improves to a 2.98% gap, surpassing all learning-based approaches. On the 1,000-customer setting, where most neural methods degrade, SEAFormer proves robust, achieving a 0.62% gap in 30 seconds, surpassing LEHD (1.05% in 2 minutes) and very recent methods such as RELD (2.36%) and L2C-Insert (0.82%). With SGBS, SEAFormer outperforms HGS by a 0.9% gap and matches UDC and BLEHD.

Moreover, we evaluate SEAFormer on benchmarks across very large-scale (Appendix C), real-world VRP, EVRPCS, and VRPTW datasets (Appendix D), and cross-distribution (Appendix E) settings. In all cases, SEAFormer performs well, establishing itself as a robust solution for research benchmarks and real-world applications.

## 5.3 ABLATION STUDY

Figure 4 shows that both EAM and CPA are essential for SEAFormer's performance. SEAFormer converges faster than POMO, whereas removing EAM results in noticeably slower convergence. This slowdown occurs because clustering modules such as CPA partition the attention space, which naturally reduces convergence speed. SEAFormer-NoCPA uses POMO's general encoder and therefore converges faster than SEAFormer-NoEAM, reaching performance close to POMO, indicating

Table 2: Performance comparison of various methods on 100 VRP instances with 100, 500, and 1,000 customers. Best values are bolded while the best learning-based solutions are highlighted.

| METHODS | 100 CUSTOMERS | | | 500 CUSTOMERS | | | 1K CUSTOMERS | | |
|---|---|---|---|---|---|---|---|---|---|
| | OBJ.↓ | G(%) | T | OBJ.↓ | G(%) | T | OBJ.↓ | G(%) | T |
| HGS-PYVRP | **15.5** | **0.00** | 40M | **36.84** | **0.00** | 4H | 43.5 | 0.00 | 8H |
| POMO | 15.72 | 1.41 | 5S | 44.8 | 21.6 | 20S | 101 | 132 | 3M |
| GLOP-LKH3 | 21.3 | 36.5 | 30S | 42.45 | 15.22 | 3M | 45.9 | 5.51 | 2M |
| DEEPMDV-LKH3 | 16.2 | 4.51 | 90S | 40.2 | 9.12 | 4M | 45.0 | 3.44 | 8M |
| ELG | 15.8 | 1.93 | 30S | 38.34 | 4.07 | 2.6M | 43.58 | 0.18 | 15M |
| UNITEFORMER | 15.74 | 1.54 | 5S | 41.2 | 11.8 | 3M | 61.8 | 42.0 | 22M |
| EFORMER | 15.77 | 1.74 | 6S | 46.8 | 27.03 | 3.3M | 87.4 | 100 | 27M |
| DAR | - | - | - | 38.21 | 3.7 | 30S | 43.82 | 0.73 | 3M |
| LEHD | 16.2 | 4.51 | 5S | 38.41 | 4.26 | 20S | 43.96 | 1.05 | 2M |
| BLEHD-PRC50 | - | - | - | 41.50 | 12.64 | 2.5M | 43.13 | -0.8 | 6M |
| RELD | 15.75 | 1.61 | 5S | 38.33 | 4.04 | 30S | 44.53 | 2.36 | 50S |
| L2C-INSERT | 15.72 | 1.41 | 2M | 38.72 | 5.1 | 7M | 43.86 | 0.82 | 13M |
| UDC$_{50}(\alpha = 50)$ | - | - | - | 38.34 | 4.07 | 7M | 43.48 | 0.00 | 14M |
| UDC$_{250}(\alpha = 50)$ | - | - | - | 37.99 | 3.12 | 30M | **43.00** | **-1.1** | 1.2H |
| SEAFORMER | 15.82 | 2.06 | 5S | 38.55 | 4.64 | 15S | 43.77 | 0.62 | 30S |
| SEAFORMER-SGBS | 15.72 | 1.41 | 30S | 37.94 | 2.98 | 12M | 43.10 | -0.9 | 1.5H |

that EAM combined with a general encoder can perform slightly better than models without EAM. Together, CPA and EAM create a complementary effect: CPA provides scalability through its $O(n)$ clustering mechanism and geometrically informed attention, while EAM delivers edge guidance that accelerate learning. Neither component achieves these advantages alone.

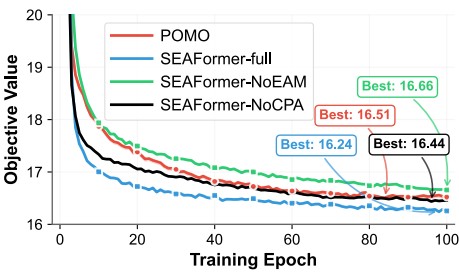

Figure 4: Training curves on 100-customer VRP (NoCPA/NoEAM = Without CPA/EAM).

Table 3: Performance comparison of CPA versus alternative clustering approaches. Attention is applied within each cluster while keeping all other parameters in SEAFormer fixed. Gaps are w.r.t full SEAFormer with CPA clustering.

| Method | VRP100 | VRP500 | VRP1000 |
|---|---|---|---|
| Reformer-LSH4 | 0.3% | 0.54% | 1.27% |
| K-Means (one round) | 7.8% | 17.2% | 28.7% |
| K-Means (four round) | 2.6% | 3.4% | 3.9% |
| Grid-based clustering | 8.14% | 15.76% | 24.05% |

To validate CPA's effectiveness, we replace it with alternative clustering strategies in SEAFormer (Table 3). Reformer with LSH4 yields the smallest gap (0.3–1.27%), showing hash-based clustering can be competitive, though CPA remains superior. Multi-round K-Means improves over single-round, reducing the gap from 28.7% to 3.9% on VRP1000, underscoring the importance of iterative refinement. Grid-based clustering performs poorly, with up to 24.05% gap on VRP1000. These results confirm CPA's advantage for routing. For a more extensive ablation study, see Appendix F.

## 6 CONCLUSION AND FUTURE WORK

In this paper, we introduce SEAFormer, a scalable, edge-aware transformer for VRP and RWVRP variants that combines Clustered Proximity Attention with a lightweight edge module. SEAFormer outperforms state-of-the-art RWVRP methods by at least 15% on large-scale instances, converges rapidly, and achieves strong performance on standard VRPs. Although the current approach has limitations, such as its single-depot focus and the training cost associated with handling different VRP variants, future work may explore multi-task learning across RWVRPs to develop a unified model trained once, extensions to the MDVRP setting, and heatmap-guided search strategies beyond greedy decoding to further leverage our edge-aware representations.

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

# A  PROBLEM FORMULATIONS OF RWVRPS

## A.1  VEHICLE ROUTING PROBLEM WITH REPLENISHMENT STOPS (VRP-RS)

The Vehicle Routing Problem with Replenishment Stops (VRP-RS) (Schneider et al. 2015) extends the traditional VRP by incorporating replenishment stops where vehicles can restock goods at intermediate nodes to continue their service. This model accounts for real-world constraints such as maximum driving range limitations due to driver working hours or operational constraints.

Let $\mathcal{V}$ define a set of vehicles, where each vehicle $w \in \mathcal{V}$ has a capacity $Q_w$. Each customer $n \in \mathcal{N}$ has a demand $q_n$ and $c \in \mathcal{C}$ shows a set of intermediate replenishment stops, which is a subset of $U = \{n_0, n_1, ..., c_1, c_2\}$. The distance between any two nodes $i$ and $j$, where $i, j \in U$, is shown as $d_{ij}$. $Q$ defines the maximum vehicle capacity, while the maximum driving range is indicated by $R$.

**Decision variables:**

- $x_{ij}^w \in \{0, 1\}$: A binary variable that takes the value 1 if vehicle $w$ travels from node $i$ to node $j$, and 0 otherwise.
- $q_i^w$: the load at node $i$ delivered by vehicle $w$
- $r_i^w$: the remaining driving range of vehicle $w$ after visiting node $i$

**Problem formulation:**

$$\text{Minimize:} \sum_{w \in V} \sum_{i \in U} \sum_{j \in U} d_{ij} x_{ij}^w \tag{11}$$

Where,

All vehicles start and end at the depot:

$$\sum_{j \in U} x_{0j}^w = 1, \quad \sum_{i \in U} x_{i0}^w = 1, \quad \forall w \in \mathcal{V} \tag{12}$$

Every customer must be visited exactly once:

$$\sum_{w \in \mathcal{V}} \sum_{j \in U} x_{ij}^w = 1, \quad \forall i \in \mathcal{N} \tag{13}$$

At each node, the number of incoming and outgoing flows must match:

$$\sum_{i \in U} x_{ij}^w = \sum_{k \in U} x_{jk}^w, \quad \forall j \in U, \forall w \in \mathcal{V} \tag{14}$$

The capacity limit of each vehicle must be satisfied:

$$q_j^w \le Q, \quad \forall j \in \mathcal{N}, \forall w \in \mathcal{V} \tag{15}$$

A vehicle's remaining capacity decreases after serving a customer:

$$q_j^w = q_i^w - q_j x_{ij}^w, \quad \forall i, j \in \mathcal{N}, \forall w \in \mathcal{V} \tag{16}$$

The total travel time of each vehicle must not exceed its maximum driving range:

$$0 \le r_i^w \le R, \quad \forall i \in U, \forall w \in \mathcal{V} \tag{17}$$

The remaining driving range is updated after each traversal:

$$r_j^w = r_i^w - d_{ij} x_{ij}^w, \quad \forall i, j \in U, \forall w \in \mathcal{V} \tag{18}$$

Traveling from the current node to a customer and then to the depot must not exceed the vehicle's driving range:

$$r_i^w \ge d_{ij} + d_{j0}, \quad \forall i, j \in U, \forall w \in \mathcal{V} \tag{19}$$

Visiting a replenishment stop fully restores the vehicle's capacity:

$$q_j^w = Q, \quad \forall j \in C, \forall w \in \mathcal{V} \text{ such that } \sum_{i \in U} x_{ij}^w = 1 \tag{20}$$

## A.2 Electric Vehicle Routing Problem with Charging Stations (EVRP-CS)

The Electric Vehicle Routing Problem with Charging Stations (EVRP-CS) (Koç et al., 2019) extends the VRP by adding a fleet of EVs that have limited driving ranges due to battery constraints. There is also set of charging stations along the routes to maintain operational feasibility.

We use the same notation as VRP-RS for problem formulation, with the key difference being that intermediate stops $c \in \mathcal{C}$ now represent charging stations rather than replenishment stops. EVs must visit these charging stations to recharge their batteries and continue their routes.

**Decision variables:**

- $x_{ij}^w \in \{0, 1\}$: binary variable that equals 1 if vehicle $w$ travels from $i$ to $j$, and 0 otherwise
- $q_i^w$: the load at node $i$ for vehicle $w$
- $r_i^w$: the remaining battery range of vehicle $w$ at node $i$

**Problem formulation:**

$$\text{Minimize: } \sum_{w \in V} \sum_{i \in U} \sum_{j \in U} d_{ij} x_{ij}^w \tag{21}$$

Where,

All vehicles start and end at the depot:

$$\sum_{j \in U} x_{0j}^w = 1, \quad \sum_{i \in U} x_{i0}^w = 1, \quad \forall w \in \mathcal{V} \tag{22}$$

Every customer must be visited exactly once:

$$\sum_{w \in \mathcal{V}} \sum_{j \in U} x_{ij}^w = 1, \quad \forall i \in \mathcal{N} \tag{23}$$

At each node, the number of incoming and outgoing flows must match:

$$\sum_{i \in U} x_{ij}^w = \sum_{k \in U} x_{jk}^w, \quad \forall j \in U, \forall w \in \mathcal{V} \tag{24}$$

The capacity limit of each vehicle must be satisfied:

$$q_j^w \leq Q, \quad \forall j \in \mathcal{N}, \forall w \in \mathcal{V} \tag{25}$$

A vehicle's remaining capacity decreases after serving a customer:

$$q_j^w = q_i^w - q_j x_{ij}^w, \quad \forall i, j \in \mathcal{N}, \forall w \in \mathcal{V} \tag{26}$$

Each vehicle's battery has a minimum and maximum capacity:

$$0 \leq r_i^w \leq R, \quad \forall i \in U, \forall w \in \mathcal{V} \tag{27}$$

The level of remaining battery is updated after traversing:

$$r_j^w = r_i^w - d_{ij} x_{ij}^w, \quad \forall i, j \in U, \forall w \in \mathcal{V} \tag{28}$$

Traveling from the current node to a customer and then to the depot must not deplete the vehicle's battery:

$$r_i^w \geq d_{ij} + d_{jc}, \quad \forall i, j \in U, \forall w \in \mathcal{V}, \forall c \in \mathcal{C} \tag{29}$$

Whenever an EV visits a charging station, its battery is fully recharged:

$$r_j^w = R, \quad \forall j \in C, \forall w \in \mathcal{V} \text{ such that } \sum_{i \in U} x_{ij}^w = 1 \tag{30}$$

### A.3 VEHICLE ROUTING PROBLEM WITH TIME WINDOWS (VRPTW)

The Vehicle Routing Problem with Time Windows (VRPTW) (Kallehauge et al., 2005) extends VRP by incorporating time constraints at customer locations. Each customer must be served within a specified time window, making the problem more realistic for applications such as delivery services, waste collection, and appointment scheduling where timing is crucial.

We use the same notation as VRP-RS for problem formulation, with the key difference being that each demand $q_i$ must be served within a time window $[e_i, l_i]$, where $e_i$ is the earliest service time and $l_i$ is the latest service time. The distance between any two nodes $i$ and $j$ is denoted as $d_{ij}$, with an associated travel time $t_{ij}$. Each customer $i$ requires a service time $s_i$.

**Decision variables:**

- $x_{ij}^w \in \{0, 1\}$: binary variable that equals 1 if vehicle $w$ travels from node $i$ to node $j$, and 0 otherwise
- $T_i^w$: arrival time of vehicle $w$ at node $i$
- $q_i^w$: cumulative load of vehicle $w$ after serving node $i$

**Problem formulation:**

$$\text{Minimize: } \sum_{w \in \mathcal{V}} \sum_{i \in U} \sum_{j \in U} d_{ij} x_{ij}^w \tag{31}$$

Where,

All vehicles start and end at the depot:

$$\sum_{j \in \mathcal{N}} x_{0j}^w = 1, \quad \sum_{i \in \mathcal{N}} x_{i0}^w = 1, \quad \forall w \in \mathcal{V} \tag{32}$$

Every customer must be visited exactly once:

$$\sum_{w \in \mathcal{V}} \sum_{j \in U} x_{ij}^w = 1, \quad \forall i \in \mathcal{N} \tag{33}$$

At each node, the number of incoming and outgoing flows must match:

$$\sum_{i \in U} x_{ij}^w = \sum_{k \in U} x_{jk}^w, \quad \forall j \in \mathcal{N}, \ \forall w \in \mathcal{V} \tag{34}$$

The capacity limit of each vehicle must be satisfied:

$$q_j^w = q_i^w + q_j x_{ij}^w, \quad \forall i, j \in U, \ \forall w \in \mathcal{V} \tag{35}$$

$$q_i^w \leq Q, \quad \forall i \in U, \ \forall w \in \mathcal{V} \tag{36}$$

Time window constraints must be satisfied:

$$e_i \leq T_i^w \leq l_i, \quad \forall i \in U, \ \forall w \in \mathcal{V} \tag{37}$$

Time consistency constraints must be satisfied:

$$T_j^w \geq T_i^w + s_i + t_{ij} - M(1 - x_{ij}^w), \quad \forall i, j \in U, \ \forall w \in \mathcal{V} \tag{38}$$

where $M$ is a sufficiently large constant.

Depot time window is between the range:

$$e_0 \leq T_0^w \leq l_0, \quad \forall w \in \mathcal{V} \tag{39}$$

Decision variables must satisfy non-negativity:

$$x_{ij}^w \in \{0, 1\}, \quad T_i^w \geq 0, \quad q_i^w \geq 0 \tag{40}$$

## A.4 ASYMMETRIC VEHICLE ROUTING PROBLEM (AVRP)

The Asymmetric Vehicle Routing Problem (AVRP) (Toth & Vigo, 1999) extends the VRP by allowing the travel cost or distance from node $i$ to $j$ to differ from that of traveling from $j$ to $i$. This asymmetry reflects real-world urban transportation scenarios, where factors such as traffic congestion, one-way streets, and temporarily closed roads can cause travel times between two nodes to vary depending on the direction.

The key distinction of AVRP with other extensions is that the distance matrix is asymmetric: $d_{ij} \neq d_{ji}$ for some pairs $(i, j)$. This creates a directed graph $G = (U, A)$ where $A$ is the set of directed arcs.

**Decision variables:**

- $x_{ij}^w \in \{0, 1\}$: binary variable that equals 1 if vehicle $w$ traverses arc $(i, j)$
- $q_i^w$: load of vehicle $w$ after serving node $i$

**Problem formulation:**

$$\text{Minimize:} \sum_{w \in \mathcal{V}} \sum_{(i,j) \in A} d_{ij} x_{ij}^w \tag{41}$$

Where,

All vehicles start and end at the depot:

$$\sum_{j \in \mathcal{N}} x_{0j}^w = 1, \quad \sum_{i \in \mathcal{N}} x_{i0}^w = 1, \quad \forall w \in \mathcal{V} \tag{42}$$

Every customer visited exactly once:

$$\sum_{w \in \mathcal{V}} \sum_{i \in U, i \neq j} x_{ij}^w = 1, \quad \forall j \in \mathcal{N} \tag{43}$$

The capacity limit of each vehicle must be satisfied:

$$q_j^w = q_i^w + q_j x_{ij}^w, \quad \forall i, j \in U, \ \forall w \in \mathcal{V} \tag{44}$$

$$q_i^w \leq Q, \quad \forall i \in U, \ \forall w \in \mathcal{V} \tag{45}$$

Vehicle route continuity:

$$\sum_{j \in \mathcal{N}} x_{0j}^w \leq 1, \quad \sum_{i \in \mathcal{N}} x_{i0}^w \leq 1, \quad \forall w \in \mathcal{V} \tag{46}$$

$$\sum_{i \in U, i \neq j} x_{ij}^w = \sum_{k \in U, k \neq j} x_{jk}^w, \quad \forall j \in \mathcal{N}, \ \forall w \in \mathcal{V} \tag{47}$$

### A.4.1 DATASET GENERATION

To generate asymmetric dataset instances, we implement a directional cost perturbation approach. We randomly select $\beta$ (asymmetry scaling parameter) customers as origin nodes, then independently select $\beta$ different customers as destination nodes. For each origin-destination pair, we introduce asymmetry by augmenting the forward travel distance by a random factor uniformly sampled from $[1, 1+\gamma]$, where $\gamma = 0.2$ in our experiments, while keeping the reverse direction unchanged. This creates directional biases that mirror real-world scenarios such as one-way streets, traffic patterns, or elevation changes.

# B    PROACTIVE MASKING FUNCTION

A masking function is an essential component in approaches for VRP optimization to restrict the search space and prevent model from generating infeasible solutions. The definition of a masking function varies for each specific problem. In the EVRPCS, infeasibility is defined as selecting an already-visited customer, selecting a customer which violate cargo capacity or entails EV to be out of battery before reaching a CS. In the VRPRS, infeasibility occurs when a previously visited customer is visited again, a customer that violate cargo weight is open to select, or decisions that cause the vehicle to exceed its maximum allowable driving range before returning to the depot.

In standard VRP, at each decision step, we evaluate whether any of the remaining customers can be feasibly visited based on the vehicle's current load and remaining capacity. However, VRPRS introduces additional complexity: a vehicle may lack sufficient driving range to reach any subsequent location after completing its current service, rendering the proposed solution infeasible. Similarly, in EVRPCS, the battery constraint adds another layer of feasibility checking beyond simple capacity constraints.

Previous approaches, such as those proposed by Wang et al. (2024), have introduced penalty functions to encourage the agent to autonomously explore the feasible domain and learn to generate valid solutions. While this penalty-based approach shows promise, it significantly expands the search space, resulting in longer convergence times during model training. Furthermore, even after complete training, such models may still produce infeasible solutions, compromising their practical reliability.

To address these limitations, we propose a proactive masking function that preemptively eliminates infeasible actions from the decision space. During the decoding step at time $t$, any customer or intermediate stop is masked if: (i) the customer has already been visited up to time $t-1$, (ii) visiting the customer would cause the vehicle to exceed its capacity limit, or (iii) the vehicle's remaining battery charge (for EVRPCS) or driving range (for VRPRS) is insufficient to reach the selected node and subsequently travel to the nearest charging station, replenishment stop, or depot.

This prevents the model from making locally feasible but globally infeasible decisions. While MT-POMO, MVMoE, and RouteFinder consider resource constraints for VRPL (vehicle routing with limited resources), they handle infeasibility through penalty functions during training rather than hard masking, meaning they can still generate infeasible solutions that require costly post-processing repair. In contrast, our proactive masking guarantees 100% feasibility by construction during inference, eliminating repair overhead and forcing the model to focus solely on viable routing decisions rather than relying on penalty-based correction.

## B.1    PROACTIVE MASKING FUNCTION FOR EVRPCS

For the EVRPCS, let $D = \{n_0\} \cup \mathcal{C}$ define the set containing both the depot and all charging stations, where $\mathcal{C}$ represents the set of charging stations. Let $\wedge$ denote the logical AND operator, $\phi \subset \mathcal{N}$ represent the set of visited customers, and $\mathcal{U}_t(j)$ indicate the masking function for visiting node $j$ at time $t$ when vehicle $w$ is currently at node $i$. The masking function is defined as:

$$\mathcal{U}_t(j) = \begin{cases} \text{False}, & j \notin \phi \wedge q_i^w + q_j \leq Q \wedge 1131 \\ \text{True}, & \text{Otherwise} \end{cases} \tag{48}$$

This formulation ensures that a node $j$ is only selectable if: (1) it has not been visited, (2) serving it would not exceed vehicle capacity, and (3) the vehicle has sufficient battery to reach node $j$ and then travel to the nearest charging facility or depot.

## B.2    PROACTIVE MASKING FUNCTION FOR VRPRS

For the Vehicle Routing Problem with Replenishment Stops, let $\mathcal{C}$ define the set of intermediate replenishment stops, $n_0$ denote the depot node, $R$ represent the maximum driving range of the vehicle, and $\phi \subset \mathcal{N}$ show the set of visited customers. Let $r_i^w$ denote the remaining driving range of vehicle $w$ at node $i$. The lookahead masking function $\mathcal{U}_t(j)$ for VRPRS, which determines the feasibility of visiting node $j$ at time $t$ when vehicle $w$ is at node $i$, is defined as:

$$\mathcal{U}_t(j) = \begin{cases} \text{False}, & j \notin \phi \wedge q_i^w + q_j \leq Q \wedge r_i^w + d_{ij} + d_{j0} \leq R \\ \text{True}, & \text{Otherwise} \end{cases} \tag{49}$$

This ensures that a customer can only be selected if visiting them and returning to the depot would not violate the driving range constraint.

## C EXPERIMENTS ON VERY LARGE VRP PROBLEMS

We evaluated SEAFormer's scalability on the exceptionally large problem instances proposed by Hou et al. (2022), comprising 5,000 and 7,000 nodes in both VRP and RWVRP settings. The results, presented in Table 4, show that SEAFormer consistently outperforms state-of-the-art methods across all RWVRP configurations and scales. The same holds for standard VRP, where SEAFormer surpasses UDC which is the leading divide-and-conquer approach for large-scale VRPs.

Table 4: Experimental results on very large-scale VRP instances. The best overall performance is shown in bold, and the top learning-based method is shaded. We compare the best-performing baselines that run without encountering out-of-memory errors in a reasonable time. Gaps are measured with respect to the best-performing approach.

| METHODS | 5K CUSTOMERS | | | 7K CUSTOMERS | | |
|---|---|---|---|---|---|---|
| | OBJ. | G(%) | T | OBJ. | G(%) | T |
| LKH | 175 | 26.7 | 4.3H | 245 | 30.2 | 14H |
| TAM-LKH3 | 144.6 | 4.7 | 35M | 196.9 | 4.67 | 1H |
| GLOP-LKH3 | 142.4 | 3.11 | 8M | 191.2 | 1.64 | 10M |
| LEHD | 140.7 | 1.88 | 3H | - | - | - |
| UDC$_{250}(\alpha = 1)$ | 139.0 | 0.65 | 15M | 188.6 | 0.26 | 20M |
| SEAFORMER | **138.1** | **0.00** | 22M | **188.1** | **0.00** | 34M |

Table 5 presents results for very large-scale RWVRPs. SEAFormer demonstrates exceptional scalability on 5,000- and 7,000-customer instances, achieving the best solutions across all four problem variants while existing methods struggle. For VRPTW, it reduces objectives by 94.3% compared to MTPOMO on 5K instances and is the only method effectively solving 7K instances. In EVRPCS, SEAFormer outperforms specialized methods like EVGAT by 45% on 5K instances and maintains this advantage at 7K scale, with similar results for VRPRS. For AVRP, it achieves a 5.57% improvement over the state-of-the-art solver, highlighting SEAFormer's strong generalizability across different problem types and scales.

Table 5: Performance of SEAFormer on a very large scale RWVRP instances. The best results are bolded while the best learning-based method is highlighted. Gaps are measured with respect to the best-performing approach.

| RWVRP | METHODS | 5K CUSTOMERS | | | 7K CUSTOMERS | | |
|---|---|---|---|---|---|---|---|
| | | OBJ. | G(%) | T | OBJ. | G(%) | T |
| VRPTW | POMO | 997 | 67.2 | 28M | - | - | - |
| | MTPOMO | 1158 | 94.3 | 30M | 1656 | 110 | 54M |
| | SEAFORMER | **596** | **0.00** | 33M | **786** | **0.00** | 65M |
| EVRPCS | EVRPRL | 272.4 | 90.3 | 30M | - | - | - |
| | EVGAT | 207.9 | 45.2 | 38M | 296.8 | 50.5 | 75M |
| | SEAFORMER | **143.1** | **0.00** | 32M | **197.1** | **0.00** | 65M |
| VRPRS | EVRPRL | 193.2 | 64.9 | 30M | - | - | - |
| | EVGAT | 167.3 | 42.8 | 38M | 230.6 | 46.8 | 75M |
| | SEAFORMER | **117.1** | **0.00** | 32M | **157.1** | **0.00** | 65M |
| AVRP | POMO | 244 | 68 | 17M | - | - | - |
| | UDC$_{250}(\alpha = 1)$ | 154.8 | 7.05 | 14M | 208.4 | 5.57 | 20M |
| | SEAFORMER | **144.6** | **0.00** | 21M | **197.4** | **0.00** | 34M |

# D EXPERIMENTS ON REAL-WORLD DATASETS

## D.1 CVRPLIB DATASET

On large-scale CVRPLib instances, SEAFormer demonstrates strong performance. Table 7 shows the gap of different methods relative to the best-known solution. When combined with SGBS, it surpasses state-of-the-art solutions; even without SGBS, SEAFormer is only 0.2% less effective on the largest instances, underscoring the robustness of the proposed approach.

Table 6: Gap to Best Known Solution on CVRPLib real-world Benchmark. The best overall performance is highlighted.

| Dataset | GLOP-LKH3 | LEHD | UDC$_{250}$ | SEAFormer | SEAFormer-SGBS |
|---|---|---|---|---|---|
| Set-X(500, 1000) | 16.8% | 17.4% | 7.1% | 7.3% | 6.8% |
| Set-XXL(1000, 10000) | 19.1% | 22.2% | 13.2% | 13.3% | - |

## D.2 EVRPCS

We evaluate SEAFormer on the real-world EVRPCS benchmark dataset from Mavrovouniotis et al. (2020), which contains 24 instances with customer sizes ranging from 29 to 1,000 and 4–13 charging stations. We focus on the 16 larger instances with more than 100 customers to assess scalability on realistic problem sizes. Table 7 reports the gap to best-known solutions, showing that SEAFormer significantly outperforms existing learning-based methods. While EVGAT and EVRPRL reach gaps of 22.6% and 24.5%, respectively, SEAFormer with greedy decoding reduces the gap to 8.2%, and SEAFormer-SGBS further improves it to 6.5%.

Table 7: Gap to Best Known Solution on EVRPCS real-world Benchmark. The best overall performance is highlighted.

| Dataset | EVGAT | EVRPRL | SEAFormer | SEAFormer-SGBS |
|---|---|---|---|---|
| EVRPCS(100, 1000) | 22.6% | 24.5% | 8.2% | 6.5% |

## D.3 VRPTW

We evaluate SEAFormer on the VRPTW benchmark (Gehring & Homberger), which includes 60 problem instances with 1,000 customers. Table 8 shows that SEAFormer substantially outperforms existing neural baselines on these real-world large-scale instances. MTPOMO and POMO yield gaps of 56.1% and 47.3%, and the distance-aware DAR method reports a 21.4% gap. In contrast, SEAFormer with greedy decoding reduces the gap to 9.1%, and SEAFormer-SGBS attains an even smaller gap of 7.4%.

Table 8: Gap to Best Known Solution on VRPTW real-world Benchmark. The best overall performance is highlighted.

| Dataset | MTPOMO | POMO | DAR | SEAFormer | SEAFormer-SGBS |
|---|---|---|---|---|---|
| EVRPTW(1000) | 56.1% | 47.3%% | 21.4% | 9.1% | 7.4% |

# E CROSS-DISTRIBUTION GENERALIZATION

A key requirement for any neural combinatorial optimization (NCO) solver is the ability to generalize beyond the data on which it was trained (Zheng et al., 2024). To evaluate this property, we assessed SEAFormer on two challenging out-of-distribution settings proposed by Zhou et al. (2023): the Rotation and Explosion distributions of the CVRP, each with 500 and 1,000 customers. As reported in Table 9, SEAFormer demonstrates consistent robustness and strong performance even when faced with large-scale instances that exhibit fundamentally different spatial structures.

Table 9: Cross-distribution generalization on 128 instances from the dataset of Zhou et al. (2023). The best overall result is shown in bold, and the top learning-based result is highlighted.

| Method | Rotation | | | Explosion | | |
| --- | --- | --- | --- | --- | --- | --- |
| | Obj.↓ | Gap | Time | Obj.↓ | Gap | Time |
| HGS | **32.97** | 0.00% | 8h | **32.87** | 0.00% | 8h |
| POMO | 64.76 | 96.4% | 1m | 58.17 | 76.9% | 1m |
| Omni_VRP | 35.9 | 8.8% | 56.8m | 35.65 | 8.45% | 56.8m |
| ELG | 37.31 | 13.16% | 16.3m | 36.53 | 8.1% | 16.6m |
| UDC-$x_{250}(\alpha=1)$ | 35.14 | 6.58% | 3.3m | 35.11 | 6.81% | 3.3m |
| SEAFormer | 35.27 | 6.97% | 1m | 35.85 | 9.06% | 1m |
| SEAFOrmer-SGBS | 34.51 | 4.67% | 100m | 34.84 | 5.99% | 100m |

# F ADDITIONAL ABLATION STUDY

## F.1 ABLATION ON CLUSTER SIZE AND PARTITIONING ROUNDS: IMPACT ON SOLUTION QUALITY AND MEMORY

One of the main contributions of this paper is CPA with its partitioning rounds and smoothing techniques. Figure 5a demonstrates model accuracy under different partitioning and smoothing settings. The SEAFormer's performance improves with increasing partitioning rounds. The accuracy gap between utilizing CPA with 1 partitioning round (PR) with and without smoothing is 0.15% for VRP1000, validating our smoothing approach's effectiveness. Furthermore, the gap reduction from 0.38% to 0.05% demonstrates the power of our deterministic-yet-diverse partitioning strategy. Notably, performance across different partitioning rounds remains relatively consistent, highlighting SEAFormer's architectural strength which achieves high-quality solutions by deterministically attending to small node groups.

Performance gains come at the cost of higher memory usage. Figure 5b illustrates memory consumption during CVRP1k training with batch size 32 and pomo size 100. As anticipated, increasing partitioning rounds and enabling smoothing raise memory requirements, highlighting the trade-off between computational resources and solution quality.

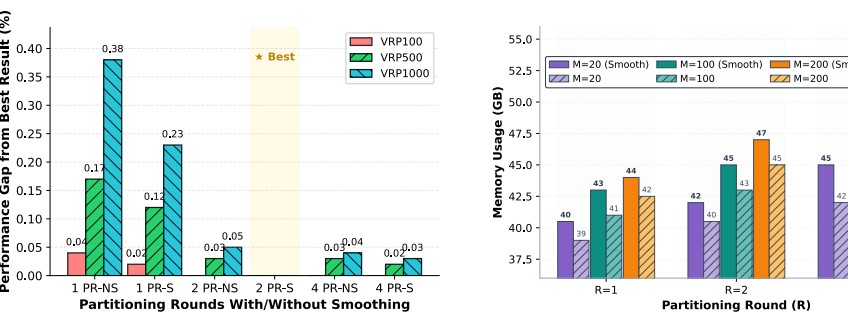

(a) Performance gap for different settings  (b) Training memory consumption

Figure 5: (a) Performance gap of SEAFormer with different CPA configurations. (b) Training memory consumption for a 1,000-node VRP with batch size 32 and pomo size of 100 under different CPA configurations.

Figure 6 compares CPA attention memory usage with POMO's standard encoder on CVRP1k in logarithmic scale. CPA achieves a 92% reduction with 20 nodes per cluster and 85% with 100 nodes, relative to full-node attention.

## F.2 ABLATION ON CLUSTER SIZE AND PARTITIONING ROUNDS: IMPACT ON CONVERGENCE

Table 10 examines the convergence behavior of different CPA configurations on VRP100. We report the gap relative to the best objective observed across the entire training process, considering two

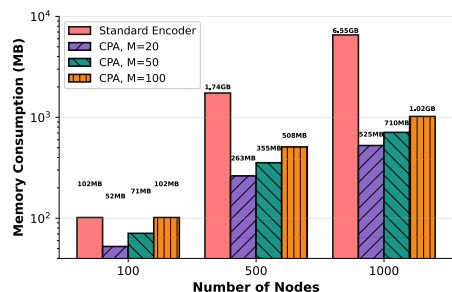

Figure 6: Logarithmic training memory consumption of CPA encoder on a 1,000-node VRP with batch size 32 and pomo size 100 under different configurations.

cluster sizes (20 and 50). In this evaluation, we vary the number of partitioning rounds (PR) and compare settings with smoothing (S) and without smoothing (NS).

The baseline POMO method achieves a 1.19% gap after 1000 epochs. In contrast, our CPA variants consistently deliver stronger performance, with several notable trends. First, the smoothing technique plays a critical role: comparing one partitioning round with and without smoothing reveals gap reductions of 0.51% and 0.45% at epoch 1000 for cluster sizes 20 and 50, respectively. Second, increasing the number of partitioning rounds leads to substantial improvements. For example, moving from one to four rounds with smoothing decreases the final gap from 1.19 to 0.69% for cluster size 20, and from 1.00% to 0.56% for cluster size 50.

We further observe that larger cluster sizes consistently outperform smaller ones across all settings. In particular, cluster size 50 with four partitioning rounds and smoothing achieves the best result, reaching a 0.56% gap. Beyond final performance, CPA also exhibits faster convergence: with four partitioning rounds and cluster size 50, the gap at epoch 100 is already 2.46%, outperforming POMO's performance at epoch 500. This combination of accelerated convergence and superior asymptotic performance highlights the importance of both multiple partitioning rounds and well-chosen cluster sizes within the CPA framework.

Table 10: Performance gap relative to best achieved objective on VRP100 across training epochs with different CPA configurations (PR: Partitioning Rounds, S: Smoothing, NS: No Smoothing)

| Method | Cluster size | epoch | | | Cluster size | epoch | | |
|--------|--------------|-------|-----|------|--------------|-------|-----|------|
|        |              | 100   | 500 | 1000 |              | 100   | 500 | 1000 |
| POMO   | -            | 4.16% | 1.89% | 1.19% | - | - | - | - |
| 1 PR-NS | 20 | 3.97% | 2.39% | 1.7% | 50 | 3.65% | 2.01% | 1.45% |
| 1 PR-S | 20 | 3.65% | 2.14% | 1.19% | 50 | 3.53% | 1.57% | 1.00% |
| 2 PR-S | 20 | 3.4% | 1.51% | 0.88% | 50 | 3.15% | 1.38% | 0.82% |
| 4 PR-S | 20 | 2.58% | 1.19% | 0.69% | 50 | 2.46% | 1.07% | 0.56% |

### F.3 ABLATION ON EDGE MODULE EFFECT ON AVRP

Table 11 reports the performance gap of SEAFormer without the edge module relative to the full SEAFormer. Across AVRP instances of varying sizes, the gap ranges from 2.4% to 3.4%, highlighting the significant contribution of the edge module to SEAFormer's performance in asymmetric routing settings.

Table 11: Gap between SEAFormer without Edge Embedding (SEAFormer-WOE) and the full SEAFormer on AVRP.

| Method | AVRP100 | AVRP500 | AVRP1000 |
|--------|---------|---------|----------|
| SEAFormer-WOE | 2.4% | 2.7% | 3.4% |

## F.4 ABLATION ON PARAMETER K IN EAM: IMPACT ON SOLUTION QUALITY AND MEMORY

We evaluate the sensitivity of SEAFormer to the number of nearest neighbors K used in the EAM, testing $K \in \{20, 50, 100, 200, 300\}$ on VRP-1000 and VRPTW-1000 benchmarks. Table 12 shows that performance improves monotonically as $K$ increases, but with strongly diminishing returns beyond K=50. For VRP-1000, increasing $K$ from 20 to 50 yields 0.34% improvement (from 43.90 to 43.75), while further increasing to 300 provides only an additional 0.32% gain (from 43.75 to 43.61). Similarly, on VRPTW-1000, the improvement from $K = 20$ to $K = 50$ is 0.39% (from 150.28 to 149.70), compared to just 0.29% from $K = 50$ to $K = 300$ (from 149.70 to 149.26). Moreover, memory consumption increases approximately linearly with $K$, from 2,054 MB at $K = 20$ to 3,326 MB at $K = 300$ (a 62% increase), primarily due to the larger edge embedding matrix stored during inference. These results demonstrate that $K = 50$ captures the majority of relevant edge information, while achieving favorable memory efficiency (only 2,255 MB, a modest 10% increase over $K = 20$). Larger $K$ values provide marginal quality improvements (¡0.3%) at significant memory and computational cost. We therefore recommend $K = 50$ as the default configuration, balancing solution quality and memory efficiency.

Table 12: Impact of edge module nearest neighbors K on solution quality and memory consumption across problem variants. Objective values and memory usage are averaged over 100 test instances with 1,000 customers, where each instance is solved individually.

| K | VRP-1000 | VRPTW-1000 | Memory (MB) |
|---|---|---|---|
| 20 | 43.90 | 150.28 | 2,054 |
| 50 | 43.75 | 149.70 | 2,255 |
| 100 | 43.67 | 149.59 | 2,495 |
| 200 | 43.62 | 149.32 | 2,870 |
| 300 | 43.61 | 149.26 | 3,326 |

## F.5 ABLATION ON CPA AND EAM: INFLUENCE ON MEMORY

We evaluate the training memory consumption of each SEAFormer component on VRP-1000 (batch size 32, POMO size 100), averaging results over 10 independent training runs. Our experiment results shown in Table 13 reveals that full SEAFormer with R=1, M=50 consumes 66 GB GPU memory, while SEAFormer-NoCPA (using standard $O(n^2)$ encoder instead of CPA) requires 75 GB, and SEAFormer-NoEAM (CPA only with R=2, M=50) consumes 60 GB. These results demonstrate that CPA provides substantial memory reduction (66 GB vs. 75 GB, a 12% saving), while the edge-aware module adds modest overhead (66 GB vs. 60 GB, a 10% increase), confirming that CPA's $O(n)$ complexity is the primary enabler of large-scale training, with the edge module providing critical performance benefits (Figure 4, Table 11) at acceptable memory cost.

Table 13: Impact of proposed architectural components on SEAFormer training memory consumption across different configurations. The batch, POMO, and problem size is set to 32, 100, and 1000, respectively.

| Configuration | SEAFormer | SEAFormer-NoCPA | SEAFormer-NOEAM |
|---|---|---|---|
| Memory | 66 GB | 75 GB | 60 GB |

## G INTEGRATING STANDARD VRP METHODS ON RWVRPs

As discussed earlier, EVRPCS and VRPRS impose state-dependent feasibility conditions such as battery level and remaining travel time, which are absent from standard VRP formulations. As a result, most existing neural or heuristic VRP solvers cannot be directly applied to EVRP-type problems, and many do not support EVRPCS or VRPRS at all.

To ensure a fair comparison, we have appled a consistent procedure to generate valid EVRPCS solutions from existing VRP methods. Specifically, we first run UDC and HGS as standard VRP solvers, ignoring battery constraints. We then identify any route where a vehicle would run out of battery and apply a simple repair mechanism (UDC-R / HGS-R). For each vehicle in the UDC or HGS solution, and at each customer, we check whether it can reach the next customer and subsequently a charging

station. If both are feasible, we follow the planned route; otherwise, we redirect the vehicle to the nearest charging station before continuing. Please note that, since HGS does not natively support EVRPCS, we apply this repair strategy to make it compatible with the EVRPCS setting.

Table 14: Infeasibility rate and gap to SEAFormer of using VRP solutions on EVRPCS instances. The infeasibility rate indicates the percentage of solutions that are invalid due to battery depletion of vehicle during delivery.

| Method | Size | Infeasibility rate | Objective | Gap vs. SEAFormer |
|--------|------|--------------------|-----------|-------------------|
| HGS    | 100  | 27%                | -         | -                 |
| HGS-R  | 100  | 0% (Repaired)      | 16.38     | 0.03%             |
| HGS    | 1000 | 100%               | -         | -                 |
| HGS-R  | 1000 | 0% (Repaired)      | 51.8      | 13%               |
| UDC    | 1000 | 100%               | -         | -                 |
| UDC-R  | 1000 | 0% (Repaired)      | 50.9      | 11%               |

As shown in Table 14, VRP solvers with repair strategies yield sub-optimal solutions because they rely on greedy local insertion of charging stops, which often delays charging until it is too late and forces vehicles to take long detours for recharging before they can resume deliveries. An optimal charging decision, however, requires global awareness of the entire route, not just local context. Since VRP solvers optimize primarily for distance, they cannot account for battery-dependent feasibility when choosing the next node. When charging stations are inserted after route construction, the spatial coherence of the route is disrupted. For example, a VRP route A → B → C may become A → B → charging → C, while the optimal EVRPCS solution would be A → charging → B → C.

## H  OPTIONAL NODES AND THEIR EMBEDDINGS

SEAFormer treats optional nodes differently from customer nodes for three key reasons. First, the number of optional nodes can change across problem instances, and mixing them with customers could make the customer embeddings unstable. Second, because CPA fixes the number of customers per cluster, adding variable optional nodes would require extra padding, which wastes resources. Third, CPA groups nearby nodes together, so if optional nodes were embedded with customers, some clusters might miss important optional node information, which could hurt solution quality.

For these reasons, we apply self-attention among all optional nodes to capture their inter-relationships. These embeddings are then integrated with customer node representations through a Gumbel-Softmax (Jang et al., 2016) and learnable fusion mechanism. During training, we implement a temperature annealing schedule: $\tau$ initializes at 1.0 to encourage exploration and decays by a factor of 0.99 per epoch until reaching 0.2. Through this mechanism, customer embeddings learn to initially explore all optional facilities without bias, then progressively concentrate on the most relevant service nodes as determined by spatial proximity and emerging routing patterns.

## I  CPA PARTITIONING ROUND VISUALIZATION

Figure 7 demonstrates how CPA employs 4 rounds of partitioning with boundary smoothing to generate diverse spatial patterns. This approach preserves the global perspective of problem instances while achieving significant reductions in memory usage and computational cost.

## J  ADDITIONAL RELATED WORK

### J.1  CLASSICAL APPROACHES AND THEIR LIMITATIONS

Classical (meta-)heuristic methods for RWVRPs follow a construction-destruction-improvement paradigm. Examples include variable neighborhood search with Tabu search for load-dependent energy consumption (Schneider et al., 2014), ant colony optimization with look-ahead for EVR-PCS (Mavrovouniotis et al., 2018), simulated annealing for partial recharging (Felipe et al., 2014), and adaptive large neighborhood search with load-aware power estimation (Kancharla & Ramadurai, 2018). Although these approaches yield high-quality solutions and handle complex constraints,

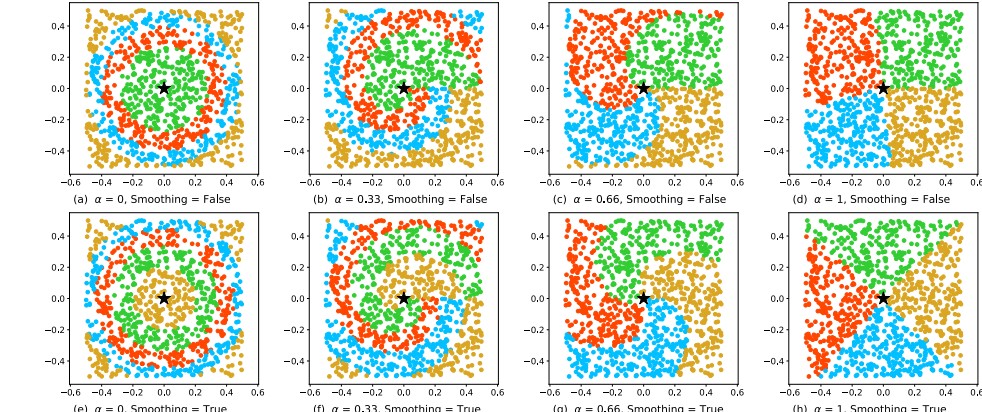

Figure 7: Visualization of Clustered Proximity Attention with $\alpha = [0, 0.33, 0.66, 1]$, where nodes are partitioned into K=4 clusters using polar coordinates centered at the depot (black star). Top row: Without boundary smoothing, hard boundaries can separate nearby nodes. Bottom row: With smoothing, cluster boundaries shift to maintain local neighborhoods. Colors indicate cluster assignment; attention is computed only within same-colored groups plus depot. The mixing parameter $\alpha$ interpolates between radial ($\alpha$=0) and angular ($\alpha$=1) clustering, capturing different routing patterns.

they face two main drawbacks: (i) solution time grows exponentially with problem size (Hou et al., 2022), and (ii) they struggle to generalize, as solutions cannot leverage patterns learned from previous instances. Such limitations make them unsuitable for real-time logistics, where solutions must be produced in seconds rather than hours.

### J.2 LEARNING-BASED APPROACHES

The intersection of machine learning and combinatorial optimization has yielded approaches that directly produce high-quality VRP solutions without iterative refinement. These methods fall into two architectural paradigms: (1) Autoregressive models that build solutions sequentially, adding one decision at a time (Khalil et al., 2017; Kool et al., 2018; Kwon et al., 2020; Hou et al., 2022; Luo et al., 2025a; 2023; Zheng et al., 2024; Berto et al., 2025; Nasehi et al., 2025), and (2) Non-autoregressive models that generate complete solutions simultaneously, typically through learned heatmaps (Nowak et al., 2018; Kool et al., 2022; Ye et al., 2024; Xiao et al., 2024). Training paradigms include supervised learning from optimal solutions or reinforcement learning to directly optimize solution quality (Joshi et al., 2019).

Early work by Bello et al. (2016) pioneered learned heuristics using pointer networks trained with actor-critic methods. Nazari et al. (2018) enhanced this framework by incorporating attention mechanisms into the encoder. The field advanced significantly when Kool et al. (2018) applied Transformers to routing problems, demonstrating strong performance across TSP and VRP tasks. Subsequent improvements include POMO (Kwon et al., 2020), which introduced multiple rollouts for better exploration, and multi-decoder architectures (Xin et al., 2021) for enhanced solution refinement. Despite these advances, scaling to large problem instances remains computationally prohibitive.

Divide-and-conquer strategies have emerged as the dominant approach for large-scale instances (Nowak et al., 2018; Li et al., 2021; Zong et al., 2022; Fu et al., 2021; Zheng et al., 2024; Nasehi et al., 2025). TAM (Hou et al., 2022) employs two-stage decomposition followed by LKH3 (Helsgaun, 2017) for sub-problem resolution. GLOP (Ye et al., 2024) integrates global partitioning through non-autoregressive models with local autoregressive construction. UDC (Zheng et al., 2024) addresses sub-optimal partitioning through robust training procedures, combining GNN-based decomposition with specialized sub-problem solvers.

Recently, heavy decoder architectures have emerged as another approach for achieving high-quality results. Luo et al. (2023) proposed shifting computational complexity from the encoder to the de-

coder, introducing a partial reconstruction approach to enhance model accuracy while maintaining reasonable training times through supervised learning. Luo et al. (2025a) extended this with a boosted heavy decoder variant that restructures attention computation through two intermediate nodes, reducing attention complexity to $O(2n)$. This efficiency gain enables training on significantly larger instances by computing attention from each node to two pivot nodes, then from these pivots to all other nodes, rather than computing full pairwise attention. Huang et al. (2025) provide insights into models with light encoders and identify their weaknesses. They propose RELD, incorporating simple modifications such as adding identity mapping and a feed-forward layer to enhance the decoder's capacity.

### J.3 LOCAL ATTENTION MECHANISM

Existing local-attention (Fang et al., 2024; Gao et al., 2024; Wang et al., 2025; Li et al., 2025; Zhou et al., 2024) approaches operate primarily in the decoder and modify attention weights based on distance, aiming to speed up decoding and improve generalization in neural VRP solutions. While these techniques can improve scalability on simple VRPs, modifying decoding scores introduces a strong locality bias that may degrade solution quality for complex RWVRPs, such as VRPTW, EVRPCS, VRPRS, and AVRP, where feasible actions often depend on long-range decisions and state-dependent constraints. In contrast, CPA preserves the decoder structure entirely and instead introduces a pattern-based inductive bias in the encoder, enabling efficient training at scale without sacrificing solution quality.

## K HYPERPARAMETERS

Following previous work (Kwon et al., 2020), we sample depot and customer coordinates uniformly from $[0, 1]^2$ space, with customer demands drawn uniformly from $\{1, ..., 10\}$. Vehicle capacities are set to 50, 100, and 200 units per Zhou et al. (2023). For each epoch, we generate 10,000 training instances on-the-fly. We train models for 2,000 epochs on 100-customer instances, 200 epochs on 500-customer instances, and 100 epochs on 1,000-customer instances for each problem variant. Training employs batch size 64 with POMO size 100 (Kwon et al., 2020). Our architecture uses a 6-layer encoder with 8 attention heads, embedding dimension 128, and feedforward dimension 512. The learning rate remains fixed at $10^{-4}$. We leverage Flash Attention (Dao et al., 2022) in our multi-head attention mechanisms. The edge module uses 32-dimensional embeddings, computing edge representations only between each node and its 50 nearest neighbors as a fixed constraint. All remaining hyperparameters, training algorithms, and loss functions follow POMO (Kwon et al., 2020) specifications.

**VRPTW.** we follow the time window generation procedure from Liu et al. (2024). Service times and time window lengths are uniformly sampled from [0.15, 0.2], representing normalized time units relative to a planning horizon of T=4.6.

**EVRPCS.** we randomly select 4 charging stations, and each EV can travel up to 2 units before needing to recharge.

**VRPRS.** we set 5 replenishment stops, with each vehicle able to travel a maximum of 4 units.

**AVRP.** the asymmetry scaling parameter $\beta$ is set to 50, 200, and 400 for 100-, 500-, and 1,000-customer problems, respectively, while the directional bias factor $\gamma$ is fixed at 0.2 for all instance sizes.

## L MODEL PARAMETERS AND TRAINING TIME

We provide a comprehensive comparison of model parameters and training costs across SEAFormer and state-of-the-art baselines in Table 15. SEAFormer-Full contains 1.36M trainable parameters, with the edge-aware module contributing only 95K parameters (7.5% increase over SEAFormer-NoEAM's 1.27M), demonstrating its lightweight design. Compared to baselines, SEAFormer maintains competitive parameter efficiency while requiring substantially less training time: it uses similar capacity to UDC (1.51M) but trains in 87 hours versus UDC's 140+ hours (excluding partitioner pretraining), and it achieves superior performance with only 21% of EFormer's parameters (6.39M) and

29% of UniteFormer's parameters (4.74M) while training 5× faster (87H vs. 390-440H). Moreover, the edge-aware module adds modest computational overhead (+10% epoch time: 55s vs. 50s for 100-node instances).

Table 15: Comparison of trainable parameters across neural VRP methods. SEAFormer achieves competitive parameter efficiency while outperforming heavier architectures.

| Method | Parameters | Total Epochs | Epoch Size | Epoch time | Total training time |
|---|---|---|---|---|---|
| SEAFormer-NoEAM | 1.27M | 2300 | 10,000 | 50s, 9M, 20 M | 80 H |
| SEAFormer-Full | 1.36M | 2300 | 10,000 | 55s, 10M, 21M | 87H |
| UDC | 1.51M | 200 | 1000 | 42 M | 140 H + Partitioner training time |
| UniteFormer | 4.74M | 1010 | 100,000 | 23 M | 390 H |
| EFormer | 6.39M | 1010 | 100,000 | 25 M | 440 H |

## M    BASELINES, CODES AND LICENSES

As noted earlier, many existing methods cannot be directly applied to all RWVRP variants. For EVRPCS, we use two learning-based approaches from the literature, following the implementation details provided in their original papers. To our knowledge, no learning-based solutions exist for VRPRS, so we adapted the EVRPCS methods, given their similarity in implementation and approach, to VRPRS and retrained them. For AVRP, we use baseline methods to generate solutions and then recompute route costs using the asymmetric distance matrix. Reported runtimes reflect only model execution, excluding post-processing or cost recalculation. We keep all hyperparameters at the defaults specified by the original authors. We also include recent concurrent works from arXiv, as well.

**OR-Tools.** We configure OR-Tools using the PATH_CHEAPEST_ARC strategy to generate initial solutions and apply GUIDED_LOCAL_SEARCH as the local search metaheuristic. The runtime is adapted to problem size, selecting from 30, 180, or 360 seconds per instances. The results reported for OR-Tools are not the optimal, as the process was stopped once the time limit was reached.

**HGS.** We use the HGS (Helsgaun, 2017) implementation in PyVRP (Wouda et al., 2024) version 0.8.2, setting the neighborhood size to 50, minimum population to 50, and generation size to 100. Runtime is adjusted by problem size, choosing 20, 120, or 240 seconds per instance. All other parameters remain at default. The results reported for HGS are not the optimal, as the process was stopped once the time limit was reached.

The available codes and their licenses for used in this work are listed in Table 16

Table 16: Resources for TSP and Optimization Problems

| Resource | Link | License |
|---|---|---|
| LKH3 (Helsgaun, 2017) | http://webhotel4.ruc.dk/keld/research/LKH-3/ | Academic research use |
| HGS (Vidal, 2022) | https://github.com/chkwon/PyHygese | MIT License |
| PyVRP (Wouda et al., 2024) | https://github.com/PyVRP/PyVRP | MIT License |
| POMO (Kwon et al., 2020) | https://github.com/yd-kwon/POMO | MIT License |
| LEHD (Luo et al., 2023) | https://github.com/CIAM-Group/NCO_code/tree/main/single_objective/LEHD | MIT License |
| BLEHD (Luo et al., 2025a) | https://github.com/CIAM-Group/SIL | MIT License |
| ELG (Gao et al., 2024) | https://github.com/lamda-bbo/ELG | MIT License |
| GLOP (Ye et al., 2024) | https://github.com/henry-yeh/GLOP | MIT License |
| DeepMDV (Naseri et al., 2025) | https://github.com/SaeedNB/DeepMDV | MIT License |
| RELD (Huang et al., 2025) | https://github.com/ziweileonhuang/reld-nco | - |
| EFormer (Meng et al., 2025) | https://github.com/Regina921/EFormer | MIT License |
| UnitFormer (Meng et al.) | https://github.com/Regina921/UniteFormer | MIT License |
| UDC (Zheng et al., 2024) | https://github.com/CIAM-Group/NCO_code/tree/main/single_objective/UDC-Large-scale-CO-master | MIT License |

## N    MDVRP EXTENSIBILITY

While our current implementation uses polar coordinates centered on a single depot, CPA can be naturally extended to Multi-depot problems (MDVRP) through several straightforward adaptations:

**i) Unified depot representation**: Designate one depot as the primary reference point and embed all customers and secondary depots based on their spatial locations relative to this main depot. This approach requires minimal modification to our current implementation and treats additional depots as special customer nodes with supply capabilities, just like the VRPRS variant with replenishment stops. **ii) Depot-specific partitioning**: Compute polar coordinates relative to each depot independently and apply CPA within each depot's customer assignment. In this variant, instead of using multiple partitioning rounds with different $\alpha$ values, each round computes customer embeddings with respect to a different depot, creating depot-specific spatial perspectives that naturally capture which customers are best served by which depot while maintaining the same computational complexity, where now corresponds to the number of depots. The key insight is that CPA exploits radial spatial structure around central points; whether that center is a single depot or multiple depots does not change the underlying geometric patterns that we show in Section 3.

## O  USE OF LARGE LANGUAGE MODELS

We use LLM assistance for grammar and presentation improvements. The original text and ideas remain the authors' work, and they take full responsibility for the content.