# OpenReview forum: "SEAFormer: A Spatial Proximity and Edge-Aware Transformer for Real-World Vehicle Routing Problems"
_ICLR.cc/2026/Conference — Submitted to ICLR 2026_

### Official Review · Reviewer_JSsB · 2025-10-21

**Soundness:** 3
**Presentation:** 3
**Contribution:** 3
**Rating:** 6
**Confidence:** 4

**Summary:**

This paper introduces SEAFormer, a scalable and edge-aware transformer designed for real-world vehicle routing problems (RWVRPs). Specifically, it presents an efficient clustered proximity attention mechanism that restricts attention computation to nodes within the same cluster. In addition, an edge-aware module is proposed to incorporate pairwise relational information during both encoding and decoding. Extensive experiments on several large-scale RWVRPs demonstrate the effectiveness and efficiency of the proposed approach.

**Strengths:**

* This paper addresses several real-world vehicle routing problems.
* The proposed method is technically sound and significantly reduces computational complexity.
* The paper is well-written and enjoyable to read.

**Weaknesses:**

* The paper format (e.g., margins) appears to be incorrect.
* The introduction does not mention improvement-based solvers, which are an important category in this domain.
* The definition of RWVRP is not clearly stated. Recent NCO approaches have considered several VRP variants [1]. Are these also classified as RWVRPs?
* The generality of SEAFormer remains unclear. For instance, real-world VRPs often do not rely on 2D Euclidean coordinates or assume a single depot. How does SEAFormer (Eq. (1)) handle such cases?
* What is the difference between the proposed proactive masking and the conventional masking used in Transformer-based solvers? The description in lines 938–943 does not make this distinction clear. For instance, models such as MTPOMO, MVMoE, and RouteFinder also consider condition (iii) when addressing VRPL. Moreover, could the authors clarify how proactive masking further enhances solution quality and reduces the search space?
* Additional baselines should be included in Table 1, such as LKH3 and HGS for VRPTW. Furthermore, are the reported neural baselines (e.g., POMO, MTPOMO, RouteFinder) retrained under the same training settings (e.g., varied problem sizes)?
* For the ablation study in Fig. 3, how does performance change when CPA is removed?
* Have the authors conducted any hyperparameter sensitivity analyses?
* A comprehensive comparison of training and inference overheads (e.g., memory consumption, training time) should be provided.
* Section 6 should include deeper discussions on the method’s limitations and potential directions for future work.
* Minor:
  * Line 218: $s_n^{\alpha} \to s_{\alpha}^n?$
  * Line 317: what does $c_t$ within $q_t$ represent? Is it the remaining vehicle capacity?
  * Line 1210: should $\alpha$ be 1 instead of 0?
  * In Table 2, “HGS” should be replaced with “HGS-PyVRP” due to differences in performance and implementation.
  * Fig. 6 is informative and should be moved to the main paper.
  * What does $B$ denote in Figs 4 and 5?

[1] Routefinder: Towards foundation models for vehicle routing problems.

----

Overall, I believe this paper makes a valuable contribution to the VRP domain by addressing several key challenges encountered in real-world VRPs. Therefore, I recommend acceptance.

**Questions:**

* Could SEAFormer handle complex constraints, as studied in [2]?
* Is it possible to train SEAFormer on the four RWVRPs in a multi-task manner?
* I did not fully understand the issue of hard cluster boundaries and how Eq. (4) addresses it. Could the authors elaborate on this point?
* What are the node and edge features used for each problem (e.g., VRPTW)? Additionally, how would the model handle inputs in the form of distance matrices rather than coordinates, as in AVRP?

[2] Learning to Handle Complex Constraints for Vehicle Routing Problems.

---

> ### Author Response · Authors · 2025-11-28
> **Response to Reviewer JSsB - Part 1**
>
> Thank you for your valuable and insightful comments. We will revise the paper accordingly to address all noted weaknesses. Specifically, we will: provide comprehensive coverage of improvement-based methods (addressing Weakness 2); more clearly articulate the distinguishing characteristics of RWVRPs (addressing Weakness 3); explicitly demonstrate SEAFormer's extensibility to MDVRP (addressing Weakness 4); include a thorough explanation of how proactive masking improves solution quality and reduces search space (addressing Weakness 5); present new experimental results comparing against HGS (addressing Weakness 6); provide complementary ablation results for SEAFormer without CPA (addressing Weakness 7); add ablation experiments examining the parameter k in the Edge-Aware Module to the Appendix (addressing Weakness 8); include detailed analysis of inference time, training time, and memory consumption (addressing Weakness 9); and expand the conclusion with more complete discussion of future research directions and SEAFormer's limitations (addressing Weakness 10).
>
> ## Response to Weakness 1:
> We will ensure the revised manuscript strictly adheres to the ICLR 2026 style guidelines, including all margin specifications and formatting requirements before resubmission.
>
> ## Response to Weakness 2:
>
> We acknowledge that while we included the improvement-based solvers like HGS as our baselines, they were not mentioned in our literature review. In the revised paper, we will add a discussion of improvement-based solvers (local search methods, tabu search approaches, etc.) in the Introduction section and expand Additional Related Work in Appendix J of the revised paper to include their role in VRP literature. We will describe that while improvement-based methods like HGS and LKH3 achieve high-quality solutions through iterative refinement, they face two key limitations: (i) extensive runtime which grows significantly with problem size (HGS requires 8 hours for VRP-1000, Table 2), and (ii) they cannot leverage patterns across problem instances, requiring full optimization from scratch for each new problem. These limitations motivate learning-based solutions such as SEAFormer that build solutions directly in seconds as a complement rather than a replacement for improvement methods, as these solutions could serve as a strong initial solution for improvement-based refinement in hybrid approaches.
>
> ## Response to Weakness 3:
> Each of the variants mentioned in our paper is individually studied in different papers in the literature, but not unifiedly studied and solved with one model before. As we are solving all these variants with common characteristics, we used the term 'RWVRP' to denote them together. We will further clarify this term and the definition in the second paragraph of the introduction. Below are the details of the defining properties of RWVRP variants.
>
> Real-World Vehicle Routing Problems (RWVRPs) are VRP variants characterized by sequence-dependent constraints with three defining properties:
>
> * **i) Local Infeasibility:** A decision's validity cannot be determined in isolation, and it depends on the complete path history. For example, whether a vehicle can serve customer j depends on accumulated battery consumption from all previous visits (EVRPCS) or elapsed time through the route (VRPTW).
>
> * **ii) State Accumulation:** Vehicle state (battery level, current time, remaining capacity) evolves dynamically through the route and is only calculable when the visitation sequence is defined. This state cannot be predicted during clustering (divide-and-conquer methods) or the prediction of edge existence in the solution (non-autoregressive methods).
>
> * **iii) Tightly Coupled Constraints:** Violating one constraint propagates through subsequent decisions. For instance, if a vehicle doesn't charge at an appropriate station in EVRPCS, it may lose the opportunity to charge later, forcing a return to the depot despite remaining delivery capacity, resulting in suboptimal solutions.
>
> In the Routefinder [1] paper that you mentioned, there are some variations of VRP, specifically: **(i)** Time Windows (VRPTW) is clearly an RWVRP as arrival times accumulate through the route and feasibility depends on complete path history; **(ii)** Open routes (OVRP) where vehicles don't return to depot changes cost structure but doesn't introduce path-dependent state, so it cannot be categorize as RWVRPs; **(iii)** Backhaul and Mixed Backhaul (VRPB, VRPMB) have pickup-delivery sequencing constraints where vehicles must complete all deliveries before pickups, creating path-dependent feasibility that qualifies them as RWVRPs; (iv) Multi-depot (MDVRP), involves depot assignment but not sequential state accumulation, so it's not an RWVRP by our definition.
> We will revise the introduction (paragraph 2) to more clearly demonstrate the characteristics of RWVRPs.
>
> ~~~
> [1] Routefinder: Towards foundation models for vehicle routing problems.
> ~~~

---

> ### Author Response · Authors · 2025-11-28
> **Response to Reviewer JSsB - Part 2**
>
> ## Response to Weakness 4:
>
> SEAFormer's architecture can handle more general settings, **(i) Non-Euclidean distances (distance matrices):** When coordinates are unavailable, or distances are non-Euclidean (as in AVRP), the edge-aware module directly uses distance matrix entries $d_{ij}$ as edge features $X^{(i,j)}_e$ in Equation 7. This is why SEAFormer achieves strong AVRP performance (Table 1: -5.2% improvement on 1000 nodes), where asymmetric distances don't correspond to Euclidean geometry, the edge module captures directional cost differences that spatial coordinates alone cannot represent. **(ii) Multi-depot problems (MDVRP):** CPA's polar coordinate mechanism can be extended through: (a) designating one depot as the primary reference point and treating other depots as special supply nodes, requiring minimal modification to our current implementation, or (b) depot-specific partitioning where each of $R$ partitioning rounds computes embeddings relative to a different depot, naturally capturing depot-customer assignments while maintaining $O(nRM) = O(n)$ complexity.
>
> We will add a discussion section in Appendix N of the revised paper to explicitly address MDVRP extensibility and outline these potential adaptation strategies as directions for future work.
>
> ## Response to Weakness 5:
>
> The conventional masking in Transformer-based solvers (POMO, AM, etc) checks conditions (i) already visited and (ii) capacity violation, which are locally verifiable constraints. Our proactive masking additionally implements condition (iii) with path-dependent lookahead: for EVRPCS, we mask customer $j$ if $r^w_i < d_{ij} + \min\limits_{c \in D} d_{jc} $ (where $r^w_i$ shows remaining battery at node $i$, $d_{ij}$ shows energy consumption to travel from $i$ to $j$ and $D$ defines a set of charging stations in Eq. 48), requiring the vehicle to have sufficient battery to reach $j$ and then reach the nearest charging station (or depot). This prevents the model from making locally feasible but globally infeasible decisions. While MTPOMO, MVMoE, and RouteFinder consider resource constraints for VRPL (vehicle routing with limited resources), they handle infeasibility through penalty functions during training rather than hard masking, meaning they can still generate infeasible solutions that require costly post-processing repair. In contrast, our proactive masking guarantees 100% feasibility by construction during inference, eliminating repair overhead.
>
> **Regarding how this enhances solution quality and reduces the search space:** proactive masking removes actions that would lead to dead ends, forcing the model to focus solely on viable routing decisions rather than relying on penalty-based correction. In the revised paper, we will update Appendix B with a comprehensive explanation of why the proactive masking function improves solution quality and reduces the search space.
>
> ## Response to Weakness 6:
>
> We have run HGS on the VRPTW dataset, and the results are shown below. These results will be added to Table 1 in the revised paper.
> | Method | VRPTW100 | VRPTW500 | VRPTW1000 |
> |:-------|:---------|:---------|:----------|
> | HGS | 26.08 (1h) | 83.8 (5h) | 142.6 (10h) |
> | SEAFormer-SGBS | 26.5 (10s) | 85.0 (13m) | 145.2 (1.6h) |
> | SEAFormer | 26.75 (5s) | 87.4 (30s) | 149.7 (1M)|
>
> Regarding neural baselines, all methods use their original published training configurations for fair comparison. POMO, MTPOMO, and RouteFinder were trained following their respective papers' hyperparameters, and we report results on identical test sets. The training dataset generation method and distributions are identical between baselines and our method, ensuring fair evaluation on the same problem distributions. We note that retraining all baselines under identical settings with SEAFormer would require months of computation and, in many cases, is infeasible as these baselines consume high memory during training, limiting their ability to scale beyond their original configurations. These training settings is already mentioned in Appendix J of the originally submitted paper.
>
> ## Response to Weakness 7:
>
> The ablation in Figure 4 of the revised paper shows SEAFormer with/without the edge-aware module versus the POMO baseline. In the revised paper, we will add a complete ablation figure including: **(i)** POMO (full $O(n^2)$ attention baseline, **(ii)** SEAFormer-NoCPA ($O(n^2)$ attention + edge module), **(iii)** SEAFormer-NoEAM (CPA only, no edge module), and **(iv)** SEAFormer-Full (complete model). Also the complementary results of that will be included in Appendix F of the revised paper. Preliminary results show that removing CPA (variant ii) improves convergence speed compared to SEAFormer, but it cannot scale to 500+ nodes due to memory constraints.

---

> ### Author Response · Authors · 2025-11-28
> **Response to Reviewer JSsB - Part 3**
>
> ## Response to Weakness 8:
>
> We provided extensive ablation studies in Appendix F (pp. 20–22) of the originally submitted manuscript, including the results for varying M and R. Below, we are providing the details of their sensitivity and effects on the performance discussed in the originally submitted paper. In addition, a new set of experiments examining the impact of these parameters on both inference and training time will be included in the revised manuscript to more clearly characterize the sensitivity of the key hyperparameters and their effects. A new set of ablation experiments on the parameter k in the Edge-Aware Module will also be added to Appendix F.4. As we are still running the experiments, the results will be added in Appendix F, and we will notify you of the new results.
>
> The sensitivity and effects for varying M are presented in Table 8, which shows that increasing the number of partitioning rounds (R) from 1 to 4 improves the optimality gap from 1.19% to 0.56% when the cluster size (M) is fixed at 50. Using four partitioning rounds with M=50 also yields 2× faster convergence compared with POMO: SEAFormer reaches POMO’s 1,000-epoch performance in only 500 epochs, under identical settings and hardware.
>
> Figure 4a further shows that performance saturates around R=2, with diminishing returns beyond this point, while Figure 4b quantifies the memory–quality trade-off: each additional round increases memory usage by approximately 5–15%, yet provides about 0.2% improvement in solution quality.
>
> Cluster size (M) is also examined in Appendix F. Figure 4b shows that increasing M from 20 to 200 raises training memory by only ~10%, while Table 8 demonstrates that increasing M from 20 to 50 improves solution quality by ~1% relative to the best achievable objective.
>
> Taken together, these results indicate that R=2 (with boundary smoothing) and M=50 provide the best balance of accuracy and memory efficiency for SEAFormer.
>
> ## Response to Weakness 9:
>
> While we report an ablation study on SEAFormer’s memory consumption under different hyperparameters and provide inference times in Tables 1 and 2 for datasets of 100 problem instances, we acknowledge that a comprehensive efficiency comparison will further strengthen the paper. In the Appendix F of the revised paper, we will add a detailed analysis comparing SEAFormer with the baselines in terms of inference time, training time, and training-time memory consumption.
>
> ## Response to Weakness 10:
>
> Thanks for the comment. In the revised paper, we will expand Section 6 to include: **(i)** the limitations of the current implementation, such as its single-depot focus, and the training cost associated with different VRP variants that SEAFormer can handle; and **(ii)** future research directions, including multi-task learning across RWVRP variants to enable a unified model with a single training process, extensions to MDVRP, and learned heatmap-based search strategies beyond greedy decoding to better exploit our edge-aware representations. This expanded discussion will provide a more balanced view of SEAFormer's contributions, clearly acknowledge its current limitations, and outline concrete paths for advancing our architectural ideas to address broader challenges in large-scale vehicle routing.
>
> ## Minor Weaknesses:
> * **Minor Weakness 1:** Yes, we acknowledge this typo and will correct it in the revised paper.
>
> * **Minor Weakness 2:** Yes, $c_t$ represents the remaining vehicle capacity.
>
> * **Minor Weakness 3:** Based on Equation 2 in the paper, setting $\alpha = 0$ yields $s_i^{(\alpha)} = r̄_i$ (pure radial clustering), meaning nodes at similar distances from the depot are grouped regardless of angle. Therefore, Figure 7a (top-left) correctly shows concentric circular clusters corresponding to $ \alpha= 0$, where each color represents nodes at similar radial distances.
>
> * **Minor Weakness 4:** Thanks for the feedback. We will update this in the revised paper.
> * **Minor Weakness 5:** We agree with your point and acknowledge that this visualization would strengthen the main paper. Due to the page limit during submission, we placed this figure in the appendix. We will relocate this figure from the appendix to the main text to incorporate the reviewer's feedback and suggested improvements.
>
> * **Minor Weakness 6:** We acknowledge this typo in Figures 4 and 5, where the cluster size parameter is incorrectly labeled, and it should be M (not B) to denote the cluster size in CPA, consistent with our notation throughout the paper. We will correct this in the revised manuscript.

---

> ### Author Response · Authors · 2025-11-28
> **Response to Reviewer JSsB - Part 4**
>
> ## Response to Question 1:
> Thank you for this insightful reference to recent work on complex constraint handling in VRPs. We note that Bi et al. (NeurIPS 2024) study complex constraints, including time windows and vessel draft constraints. SEAFormer already demonstrates the capability to handle time window and VRPTW experiments (Table 1), showing effective handling of time window constraints with service times and temporal dependencies.
>
> **Regarding vessel draft constraints specifically:** when formulated as a VRP with depot returns (vehicles can return to depot to adjust draft), SEAFormer's proactive masking function (Appendix B) can handle this directly. Similar to how we mask infeasible charging stations in EVRPCS, we would mask customers whose draft requirements cannot be satisfied given the current vessel state and available depot returns. However, when vessel draft problems are formulated as TSP variants without depot returns, determining feasibility becomes significantly more complex, and selecting node j at step t requires predicting whether the remaining unvisited nodes can form a feasible sequence given the vessel's updated draft state, which is an NP-hard problem. Bi et al.'s key contribution is a predictive masking function that learns to approximate this long-horizon feasibility checking. Incorporating such predictive masking into SEAFormer would enable full support for TSP-style draft constraints. This could be a simple extension where Bi et al.'s masking mechanism could replace our current proactive masking (Appendix B) while leveraging SEAFormer's scalable CPA and edge-aware architecture.
>
> ## Response to Question 2:
>
> Thanks for this interesting question about multi-task learning across RWVRP variants. Yes, it is possible to train SEAFormer in a multi-task manner across the four RWVRP variants, and our architecture is naturally suited for this due to its unified design. SEAFormer's core components are problem-agnostic, while variant-specific constraints are handled through: **(i)** the state parameter ξₜ in the decoder query (See Node-Guided Sequential Attention in Section 4, page 6 of the originally submitted paper), which adapts to different state representations (battery for EVRPCS, time for VRPTW, remaining driving range for VRPRS, none for AVRP and VRP), and **(ii)** the proactive masking function (Appendix B), which enforces variant-specific feasibility and ensures that the model always generates valid solutions for any variants of the problem without requiring post-processing, thereby enabling more effective training.
>
> We have not implemented multi-task training in the current work because: **(i)** our focus was on demonstrating that a single architecture can solve all variants when trained separately, establishing feasibility before exploring joint training, and **(ii)** multi-task learning introduces additional hyperparameters (e.g., task sampling weights, gradient balancing, etc.) that require extensive tuning.

---

> ### Author Response · Authors · 2025-11-28
> **Response to Reviewer JSsB - Part 5**
>
> ## Response to Question 3:
>
> The hard cluster boundary problem occurs when nodes are partitioned based solely on sorted partitioning scores without any overlap consideration. For example, suppose after sorting by score $s^(\alpha)$, we have nodes with scores [0.498, 0.499, 0.501, 0.502] and cluster size $M=2$. A hard boundary would create Cluster 1: [0.498, 0.499] and Cluster 2: [0.501, 0.502], even though nodes with scores 0.499 and 0.501 are nearly identical in spatial proximity (differing by only 0.002) and likely to be in one delivery route. This artificial separation disrupts natural customer groupings; two spatially adjacent customers might be forced into different clusters simply because they fall on opposite sides of an arbitrary threshold, preventing them from directly attending to each other during that partitioning round.
>
> Eq.4 addresses this through circular shifting: $S_\alpha' = [s_\alpha^{\lfloor M/2 \rfloor + 1}, \ldots, s_\alpha^{n}, s_\alpha^{1}, \ldots, s_\alpha^{\lfloor M/2 \rfloor}]$. By rotating the sorted sequence by half a cluster size before partitioning, we create an alternative clustering perspective where nodes that were previously separated (at positions $M$ and $M+1$, the boundary) are now placed in the middle of clusters (at positions $M/2$ and $(M/2)+1$ within their respective new clusters). Combined with multiple partitioning rounds, this ensures that even if two nearby nodes are separated in one round (e.g., $\alpha=0$ round), they will likely be grouped together in the smoothed version or in rounds with different α values. For instance, in round 1 with $\alpha=0$, nodes A and B might be split by a boundary; in the boundary-smoothed version of round 1, they might be united; and in round 2 with $\alpha=0.33$, they likely cluster together also with other nearby nodes due to different angular weighting.
>
> In Figure 7, the bottom row shows clustering with smoothing, demonstrating how cluster boundaries appear more natural and don't artificially split tight spatial groups, whereas the top row (without smoothing) sometimes separates visually proximate nodes. Table 8 shows that boundary smoothing improves performance by 0.51% on VRP-100 (1PR-NS: 1.70% gap vs. 1PR-S: 1.19% gap), showing that this simple circular shift meaningfully enhances solution quality. If the paper gets accepted, we will add Figure 7 with a zoomed-in visualization highlighting specific node pairs that benefit from smoothing into the main paper and provide a numerical example showing how Eq. 4 transforms the partitioning.
>
> ## Responses to Question 4:
>
> Node features vary by problem variant. For all variants, base features include normalized 2D coordinates $(x, y) \in [0,1]^2$ and demand $q_i \in [1,10]$ normalized by capacity Q. For VRPTW, we add time window features $[e_i, l_i]$ normalized by horizon T=4.6 and service time $s_i \in [0.15, 0.2]$. For EVRPCS, we add a binary indicator (1 for charging stations, 0 for customers). For VRPRS, we add a binary indicator for replenishment stops. For AVRP, base features are identical to VRP since asymmetry is captured in edges, not nodes. Edge features include Euclidean distance plus cosine similarity between node coordinates, which encodes the directional orientation of the edge. For symmetric problems, the distance will be set by the input matrix, not the fixed calculated Euclidean distance.
>
> **Regarding distance matrices instead of coordinates (critical for AVRP):** Our model handles this naturally through the edge-aware module. This module directly uses distance matrix entries $d_{ij}$ as the primary edge feature $X_e^{(i,j)} $  in Eq.7, with asymmetry naturally preserved since $d_{ij} ≠ d_{ji}$ produces different embeddings $H^{(i,j)}_e ≠ H^{(j,i)}_e$. This is why SEAFormer achieves strong performance on AVRP (Table 1: -5.2% improvement on AVRP-1000) where many baselines struggle as the edge module explicitly represents directional cost differences that node embeddings alone cannot capture.

---

### Official Review · Reviewer_kLd7 · 2025-10-29

**Soundness:** 3
**Presentation:** 3
**Contribution:** 3
**Rating:** 6
**Confidence:** 4

**Summary:**

This paper tackles the critical challenge of solving large-scale Real-World Vehicle Routing Problems (RWVRPs), which existing neural methods fail to address due to sequential constraints and O(n2) complexity. The authors propose SEAFormer, a novel Transformer that introduces two key innovations: 1) Clustered Proximity Attention (CPA), a domain-specific sparse attention that reduces complexity to O(n) by clustering nodes based on polar coordinates, and 2) a lightweight Edge-Aware Module that explicitly models edge-level information via a global heatmap. Extensive experiments show SEAFormer is the first neural method to effectively solve 1000+ node RWVRPs, achieving SOTA results across four variants and classic VRP, even scaling up to 7000 nodes with massive gains over recent baselines.

**Strengths:**

Important and Challenging Problem: The paper tackles the critical gap between NCO research (small VRPs) and industrial applications (large-scale RWVRPs). Solving 1000+ node RWVRPs is a major milestone.

Novel Architectural:  The CPA innovation (based on geometric priors of polar coordinates) is an insightful design. Meanwhile, decoupling node-level sequential attention from edge-level global features is an effective way to handle heterogeneous constraints.

Strong Experimental Results: Extensive experimental evaluations are conducted to demonstrate the effectiveness of the proposed solution.

The paper is well written and easy to follow.

**Weaknesses:**

1. Incomplete Complexity Analysis: The paper claims O(n) complexity (i.e., O(nRM)) in Equation (6). This conclusion relies on R and M being fixed constants. However, the authors do not discuss the boundary condition where R *M > n, in which case O(nRM) would not hold. A rigorous theoretical analysis is encouraged.

2. Insufficient Analytical Justification: The paper lacks rigorous analytical evidence to explain the source of its strong empirical performance. It remains unclear whether the "scalability" arises purely from the O(n) complexity of CPA or from a stabilizing interaction between CPA and the EAM. Consequently, the originality of the contribution is difficult to evaluate, as the authors do not convincingly demonstrate why their domain-specific “polar clustering” is essential, or whether comparable gains could be achieved using a generic O(n) sparse attention mechanism.

3. In the Related Work section, the author discusses the limitations of existing studies, but does not clearly explain how the proposed method addresses or overcomes these limitations.

4. Potential Limitation to Multi-Depot Problems: CPA's core mechanism (polar coordinates) fundamentally limits it to Single-Depot VRPs (SDVRP), making it difficult to extend to Multi-Depot (MDVRP) settings.

5. Hyperparameter Sensitivity: The appendix shows that CPA is sensitive to its key hyperparameters (R and M), which could be a barrier to practical adoption.

**Questions:**

On the contribution of the SGBS search: To enable a more direct comparison of architectural strength, could the authors report the performance of SGBS when applied to the strongest existing baselines, such as UDC or LEHD? This would clarify whether the observed improvements stem primarily from the SGBS procedure or from the proposed architecture itself.


Implementation details: The Appendix states that the edge module computes representations only for the 50 nearest neighbors. Have the authors conducted sensitivity analyses to evaluate how the choice of k = 50 affects performance and stability?

---

> ### Author Response · Authors · 2025-11-28
> **Response to Reviwer KLd7 - Part 1**
>
> Thank you for your valuable and insightful comments. We will clarify depot padding and add extended experiments on the number of partitioning rounds (addressing Weakness 1). We will conduct thorough ablations to isolate each component’s contribution under different settings (addressing Weakness 2). We will also provide a clear mapping from each identified limitation to our corresponding solution, with forward references to the relevant sections (addressing Weakness 3). Also, we will provide an adaptation strategies to address MDVRP extensibility of our model (addressing Weakness 4). Finally, we will show that our model’s hyperparameter sensitivity is small and predictable, and we will provide practical selection guidelines with simple decision rules based on available computational resources and time (addressing Weakness 5).
>
> ## Response to Weakness 1:
>
> Thank you for your valuable comment. To clarify the architecture: $R$ represents the number of complete partitioning rounds where the entire set of $n$ nodes is re-partitioned with different α values, and $M$ represents the fixed cluster size within each round. At each partitioning round, we divide all n nodes into $⌈n/M⌉$ clusters of size $M$, and crucially, when n is not perfectly divisible by $M$ (i.e., when $⌈n/M⌉ × M > n)$, we pad the final cluster with depot nodes to ensure uniform cluster size $M$ across all partitions. This padding ensures every cluster has exactly $M$ nodes and can be processed in parallel on a GPU. The complexity analysis then becomes: for each of R rounds, we have $⌈n/M⌉$ clusters each requiring $O(M^2)$ attention operations, yielding total complexity $O(R × ⌈n/M⌉ × M^2) = O(R × n × M) = O(n)$ since both $R$ and $M$ are constants independent of $n$ ($R$=4, $M$=50 in all our experiments) - as mentioned in Equation 6 of the originally submitted paper. An empirical validation in the Ablation study in Sections F.1 and F.2 confirms the linear scaling with n for different values of $R$ and $M$.
>
> In the revision, we will add an explicit statement of the padding mechanism in Section 3, clarifying that depot padding ensures well-defined cluster boundaries when n mod $M ≠ 0$. Moreover, we will experiment on the number of partitioning rounds $R \in${1,2,4}, and $M \in${20,50,100, 200}, measuring solution quality, convergence speed, memory consumption, and both training and inference time to quantify the trade-off between complexity and performance. These results will be added to Section F.5 in the appendix.
>
> ## Response to Weakness 2:
>
> Thanks for raising the comment. To demonstrate why polar clustering is important, in Section 3 of the originally submitted paper, we show that optimal tours in VRP problems typically follow one of three dominant patterns. SEAFormer leverages these patterns through CPA and incorporates a lightweight edge-aware module, enabling training on larger instances, which then leads to improved performance. Table 3 demonstrates that CPA outperforms generic sparse attention methods (Reformer-LSH by 0.3–1.27% and K-means by 3.9%), indicating that polar clustering provides benefits beyond complexity reduction. Table 9 shows that EAM contributes a 2.4–3.4% improvement on AVRP, and Figure 4 of the revised paper illustrates that CPA alone converges more slowly than POMO, whereas CPA+EAM together converge faster.
>
> In the revised paper, we will conduct comprehensive ablation experiments to isolate each component’s contribution on both small- and large-scale problems by evaluating three additional configurations, resulting in the following full set of settings: **(i)** SEAFormer-NoCPA (standard $O(n^2)$ attention + EAM) to measure EAM's standalone value, **(ii)** SEAFormer-NoEAM (CPA only, no edge embeddings) to measure polar clustering's standalone value, **(iii)** SEAFormer-GenericSparse (Reformer + EAM) to demonstrate the benefit of polar clustering over sparse attention mechanism. These results will be added to Appendix F.5 of the revised paper, and Figure 4 will be updated accordingly.

---

> ### Author Response · Authors · 2025-11-28
> **Response to Reviwer KLd7 - Part 2**
>
> ## Response to Weakness 3:
>
> We acknowledge that in Section 2 (Related Work), we identify limitations of existing methods; however, the connections to how SEAFormer addresses these limitations could be more explicit. While SEAFormer addresses each limitation but the explanations are distributed across multiple sections: **(i)** for autoregressive methods' $O(n^2)$ memory bottleneck (Section 2.1), we introduce CPA in Section 3 achieving $O(n)$ complexity through polar clustering, enabling training on large-scale problem instances where POMO fails (Table 2); **(ii)** for divide-and-conquer methods' circular dependency problem (Section 2.2), we maintain autoregressive construction with sequential state tracking while achieving scalability through efficient attention, avoiding the clustering-before-sequencing issue; **(iii)** for non-autoregressive methods' inability to handle sequence-dependent constraints (Section 2.2, paragraph 3), our decoder explicitly handle this issue through sequential decoding and tracking vehicle state (battery in EVRPCS, spent time in VRPTW, remaining driving range in VRPRS); **(iv)** for existing methods' inability to represent edge-specific attributes (Lines 62 - 65 in introduction), we introduce the edge-aware module (Section 4) that explicitly embeds pairwise relationships, enabling asymmetric cost handling and constraint modeling. In Section 2 of the revised paper, we will include an explicit mapping of each identified limitation to our corresponding solution, and add forward references to the section that addresses the issue in the paper.
>
> ## Response to Weakness 4:
>
> We appreciate this insightful observation about CPA's applicability to multi-depot scenarios. While our current implementation uses polar coordinates centered on a single depot, CPA can be naturally extended to multi-depot problems (MDVRP) through several straightforward adaptations:
>
> * Unified depot representation where we designate one depot as the primary reference point and embed all customers and secondary depots based on their spatial locations relative to this main depot. This approach requires minimal modification to our current implementation and treats additional depots as special customer nodes with supply capabilities, just like the VRPRS variant with replenishment stops.
>
> * Depot-specific partitioning, where we compute polar coordinates relative to each depot independently and apply CPA within each depot's customer assignment. In this variant, instead of using multiple partitioning rounds $R$ with different α values, each round computes customer embeddings with respect to a different depot, creating depot-specific spatial perspectives that naturally capture which customers are best served by which depot while maintaining the same computational complexity $O(nRM)$, where $R$ now corresponds to the number of depots.
>
> The key insight is that CPA exploits radial spatial structure around central points; whether that center is a single depot or multiple depots does not change the underlying geometric patterns that we show in Section 3. We acknowledge that we have not implemented or evaluated these MDVRP extensions in the current work, as our focus was on demonstrating scalability and constraint handling for single-depot RWVRPs (VRPTW, EVRPCS, VRPRS, AVRP) where existing methods fail. We will add a discussion section in the Appendix N of the revised paper to explicitly address MDVRP extensibility and outline these potential adaptation strategies as directions for future work.
>
> ## Response to Weakness 5:
>
> In Appendix F (Table 8, Figure 4) of the originally submitted paper, we show performance variations across different configurations of $R$ (partitioning rounds) and $M$ (cluster size) and emphasize that the sensitivity is minor and follows predictable patterns. Our experiments show that performance improves monotonically with $R$ until saturation on $R=2$ (Figure 4a), and larger $M$ consistently outperforms smaller $M$ ($M=50$ better than $M=20$ across all settings in Table 8), providing clear guidelines rather than unpredictable interactions. Please note that the y-axis scale in Figure 4 covers a very small range, showing only about a 0.38% performance difference when varying $R$ from 1 to 4. We will address this concern by adding a more detailed ablation study in Appendix F.4, F.5 of the revised paper, providing decision rules based on available computational budget (e.g., for memory-constrained settings, use lower values for M and R; for quality-critical applications, use a larger value of M; for faster convergence, use larger $R$).

---

> ### Author Response · Authors · 2025-11-28
> **Response to Reviewer KLd7 - Part 3**
>
> ## Response to Question 1:
>
> Thank you for this insightful question. We have included SGBS results to demonstrate the quality achievable with our learned embeddings when given additional computational budget and time, but our core architectural claims are based on greedy decoding results. For instance, SEAFormer achieves a 0.62% gap on VRP-1000 (greedy, 30 seconds) compared to LEHD's 1.05% gap (2 minutes) and UDC's 0.00% gap (14 minutes for the $\alpha=50$ variant), showing that even without SGBS, our architecture provides competitive quality with significantly faster inference.
>
> Regarding applying SGBS to baselines like UDC or LEHD, we need to clarify that these methods already employ their own specialized search procedures that are integral to their architectures. UDC and LEHD both use random reconstruction methods, and without these procedures, their performance degrades significantly in greedy mode. For instance, UDC on VRP-1000 achieves 43.0 using their reconstruction method, but achieves 46.09 using greedy search. This demonstrates that their performance is fundamentally dependent on their specific search strategies, making direct SGBS application inappropriate. We would like to perform the random reconstruction procedure on SEAFormer to provide a more direct comparison; however, due to the limited time of the rebuttal period, implementing and properly tuning this alternative search method cannot be completed.
>
> SEAFormer's key strength lies in its versatility: while methods like UDC and LEHD achieve strong performance on standard VRPs, they cannot effectively handle RWVRPs with sequence-dependent constraints due to fundamental architectural limitations (circular dependency for divide-and-conquer, supervised learning requirements for heavy decoders). In contrast, SEAFormer maintains competitive performance on classical VRP benchmarks  (Table 2) while simultaneously achieving state-of-the-art results across four RWVRP variants (15%+ improvements on VRPTW/EVRPCS/VRPRS/AVRP at 1000-node scale, Table 1), demonstrating true generalization within a single unified architecture.
>
> ## Response to Question 2:
>
> While we chose the value of $k$ as 50 based on its effect on performance, the sensitivity analysis on this hyperparameter was not included in the original submission. In the revised paper, we will include a comprehensive set of experiments evaluating K with different values ($K \in${ 20, 50,100, 200}) across various problem sizes to rigorously assess how this choice affects both solution quality and computational efficiency. These experiments will measure: (i) objective value and gap to best-known solutions as a function of K, (ii) training time and memory consumption trade-offs, (iii) convergence speed, and (iv) inference time per instance. These results will be included in Appendix F of the revised paper and also posted here.

---

### Official Review · Reviewer_55JA · 2025-10-30

**Soundness:** 1
**Presentation:** 2
**Contribution:** 2
**Rating:** 2
**Confidence:** 4

**Summary:**

The paper introduces SEAFormer, a transformer architecture designed to address real-world vehicle routing problems (RWVRPs) by incorporating two key innovations: Clustered Proximity Attention (CPA) and an edge-aware module. The method is shown to outperform existing approaches in several benchmarks, but there are some aspects of the method’s novelty, experimental design, and theoretical foundation that could be further clarified or improved.

**Strengths:**

1. SEAFormer demonstrates competitive performance across multiple RWVRP variants.
2. The introduction of Clustered Proximity Attention (CPA) effectively reduces the computational complexity of traditional attention mechanisms.
3. The edge-aware module provides a practical solution for incorporating edge-level information, enhancing model accuracy and convergence speed.

**Weaknesses:**

1. The definition of “real-world VRP” is broad, and it is unclear what specific challenges are being addressed. Although variants like VRPTW and EVRPCS are mentioned, the paper lacks a clear explanation of why existing methods cannot be extended to these variants. The motivations behind CPA and the edge-aware module are more engineering-oriented, with little exploration of the underlying theoretical mechanisms.
2. The CPA approach lacks clear innovation when compared to existing local attention or sparse attention mechanisms. The paper should clarify how CPA offers distinct advantages over these existing approaches.
3. The paper does not sufficiently distinguish its contributions from previous work that modifies attention mechanisms with distance (such as [1-5]). A clearer explanation of how SEAFormer differs would strengthen the novelty claim.
4. The statement “SEAFormer is the first unified approach to address this comprehensive range of real-world routing constraints at scales within a single architecture” seems exaggerated, as there are other works that have addressed real-world VRPs, and it is unclear what makes this method the “first”.
5. The deterministic multi-round partitioning and boundary smoothing in CPA seem empirical, without solid theoretical justification or quantitative analysis on the trade-off between complexity and performance.
6. The edge embedding module only works for certain edges (depot-customer), raising concerns about potential information loss across the full graph. The paper should explore whether this limitation affects overall solution quality.
7. Some RWVRP variants, like EVRPCS and VRPRS, are compared with a limited number of learning-based methods. In addition, stronger baseline traditional algorithms, such as HGS, are not considered in many variants.
8. There is no in-depth comparison with existing methods on model parameters, training time, or inference time, which are critical for real-world applications.
9. If SEAFormer claims applicability to “real-world” scenarios, the paper should include experiments on real-world datasets to substantiate this claim.

```
[1] INViT: A generalizable routing problem solver with invariant nested view transformer. ICML, 2024.
[2] Towards generalizable neural solvers for vehicle routing problems via ensemble with transferrable local policy. IJCAI, 2024.
[3] Distance-aware attention reshaping for enhancing generalization of neural solvers. IEEE TNNLS, 2025.
[4] Learning to solving vehicle routing problems via local–global feature fusion transformer. Complex & Intelligent Systems, 2025.
[5] Instance-conditioned adaptation for large-scale generalization of neural routing solver. arXiv, 2025.
```

**Questions:**

Please refer to the weaknesses.

---

> ### Author Response · Authors · 2025-11-28
> **Response to Reviewer 55JA - Part 1**
>
> Thank you for your valuable feedback. To address your comments, we will revise parts of the introduction and related work to more clearly define RWVRPs, articulate our contributions, and highlight distinctions from existing solutions (weaknesses 1-4). We will also provide a new set of experiments and analyses to justify our design choices and quantify the contribution of each component to SEAFormer’s overall performance (weaknesses 5, 6, and 8). In addition, we will clarify why some RWVRP variants are compared against a limited set of baselines, and include new results using HGS and UDC with a repair strategy to show their limitations (weakness 7). Finally, we will report results on two real-world datasets for VRPTW and EVRPCS (weakness 9).
>
> ## Response to Weakness 1:
>
> Real-World Vehicle Routing Problems (RWVRPs) are VRP variants characterized by sequence-dependent constraints that are frequently encountered in real-world deployment for delivering goods to customers from a depot. Each of the variants mentioned in our paper is individually studied in different papers in the literature, but not unifiedly studied and solved with one model before. As we are solving all these variants with common characteristics, we used the term 'RWVRP' to denote them together. We will further clarify this term and the definition in paragraph 2 of the introduction. The RWVRP variants have the following three defining properties:
>
> * **i) Local Infeasibility:** A decision's validity cannot be determined in isolation, and it depends on the complete path history. For example, whether a vehicle can serve customer j depends on accumulated battery consumption from all previous visits (EVRPCS) or elapsed time through the route (VRPTW).
>
> * **ii) State Accumulation:** Vehicle state (battery level, current time, etc.) evolves dynamically through the route and is only calculable when the visitation sequence is defined. This state cannot be predicted during clustering (divide-and-conquer methods) or the prediction of edge existence in the solution (non-autoregressive methods).
>
> * **iii) Tightly Coupled Constraints:** Violating one constraint propagates through subsequent decisions. For instance, if a vehicle doesn't charge at an appropriate station in EVRPCS, it may lose the opportunity to charge later, forcing a return to the depot despite remaining delivery capacity, resulting in suboptimal solutions.
>
> We have mentioned the limitations of existing solutions on RWVRPs in the Introduction and Related Work of the original paper, but we will further clarify them in the revision. The details are as below:
>
> * **Autoregressive methods (POMO, etc., lines 56–65, 125-129):** These approaches face multiple constraints that make them unsuitable for large-scale RWVRPs. Their $O(n^2d)$ memory complexity limits training to ~200 nodes, with 1000-node instances causing out-of-memory errors on a single GPU with 80 GB memory. They treat all node pairs uniformly, ignoring the depot-centric geometric structure. Moreover, these methods cannot represent edge-specific attributes such as asymmetric costs and fail to generalize to larger instances. Table 1 shows that POMO achieves a 27.6% gap on VRPTW-1000 and struggles on other RWVRPs at scale.
>
> * **Divide-and-Conquer methods (UDC, etc., lines 66–69, 133–139):** These methods first cluster nodes before determining visit sequences. However, feasibility depends on sequence-dependent states (e.g., battery level in EVRPCS, elapsed time in VRPTW), which are unavailable during clustering. This circular dependency can lead to constraint violations once sequences are generated. While UDC works on AVRP (no state dependency), it cannot be adapted to VRPTW or EVRPCS due to this fundamental incompatibility.
>
> * **Non-autoregressive methods (lines 73–79, 149-152):** These face a similar challenge to previous approaches. They cannot ensure constraint satisfaction without traversal sequence information. Critical constraints like battery levels are path-dependent and require sequential state accumulation for feasibility.
>
> * **Supervised methods (LEHD, etc., lines 69–72, 144–147):** These rely on pre-computed optimal solutions for training. For RWVRPs, generating such data is prohibitively expensive (solving a 100-node EVRPCS to optimality may take days).
> Distance-penalty methods (ELG, etc., lines 69-72, 140-144): These restrict attention to nearby nodes using soft distance penalties, while heavy local attention in the decoder leads to $O(n^2)$ complexity. Reducing attention to distant nodes can miss critical locations (e.g., far charging stations in EVRPCS). Additionally, their running time grows significantly with the number of customers, making them impractical for large-scale RWVRPs.
>
> We will revise the introduction (paragraphs 2 and 4 of the Introduction of the originally submitted paper) and the related work (Sections 2.1 and 2.2) to more clearly show the characteristics of RWVRPs and the limitations of existing methods.

---

> ### Author Response · Authors · 2025-11-28
> **Response to Reviewer 55JA - Part 2**
>
> ## Response to Weakness 2:
> Please refer to the response of Weakness 3.
>
> ## Response to Weakness 3:
>
> Our key distinction from prior distance-based attention mechanisms [1–5] lies in how spatial information is integrated into the model. Existing methods operate mainly in the decoder, adjusting attention weights using pairwise distances and user-defined rules as soft penalties. In contrast, CPA acts in the encoder, restructuring the attention computation by embedding topological information directly into the node representations. This shift allows CPA to influence the model’s representational structure rather than only modulating attention scores during decoding. We will include these clarifications succinctly in Section 2 of the revised paper, with additional details provided in Appendix J, to further strengthen the discussion on the differences discussed in Related Work of the originally submitted paper.
>
> To illustrate the motivation behind CPA’s pattern-based clustering and its difference with the attention mechanism of the existing studies, Figure 1 shows that near-optimal VRP solutions exhibit three consistent spatial patterns relative to the depot: (i) nodes with similar angles but different distances, (ii) nodes with similar distances but different angles, and (iii) nodes in close spatial proximity. SEAFormer explicitly captures these patterns in its encoder through our Clustered Pattern Attention (CPA) mechanism, enabling a pattern-based encoding structure that substantially reduces memory consumption during both training and inference while improving generalization to large-scale instances. Whereas standard attention requires $O(n^2)$ complexity per layer, CPA reduces encoder complexity to $O(R×⌈n/M⌉×M^2)= O(nRM)= O(n)$ since R and M are constants (Eq. 6). This results in a significant memory reduction in the encoder, as shown in Figure 5, which enables training on larger instances and leads to improved performance.
>
> Existing local-attention [1-5] approaches operate primarily in the decoder and modify attention weights based on distance, aiming to speed up decoding and improve generalization in neural VRP solutions. While these techniques can improve scalability on simple VRPs, modifying decoding scores introduces a strong locality bias that may degrade solution quality for complex RWVRPs, such as VRPTW, EVRPCS, VRPRS, and AVRP, where feasible actions often depend on long-range decisions and state-dependent constraints. In contrast, CPA preserves the decoder structure entirely and instead introduces a pattern-based inductive bias in the encoder, enabling efficient training at scale without sacrificing solution quality.
>
> Our investigation shows that DAR [3] is the only method among these with an implementation for VRPTW, and it is currently one of the few approaches capable of solving VRPTW at both small and large scales. We will include a full set of comparative experiments on our evaluation datasets for both VRP and VRPTW in Tables 1 and 2 of the revised paper, considering DAR as a baseline. Preliminary results indicate that the performance gap between DAR and SEAFormer is small on standard VRP, but significantly larger on VRPTW, where SEAFormer achieves much stronger performance. The results on the real-world dataset presented for Weakness 9 also show that SEAFormer outperforms DAR on real-world VRPTW instances.
>
> Our findings in comparison with baselines in Tables 1 and 2 support our claim that modifying decoder-level distance-based attention can overlook globally optimal actions and lead to lower-quality solutions in complex RWVRPs, whereas SEAFormer’s encoder-level pattern-based embedding with edge-module avoids this issue while maintaining strong generalization and high solution quality.
>
> ~~~
> [1] INViT: A generalizable routing problem solver with invariant nested view transformer. ICML, 2024.
> [2] Towards generalizable neural solvers for vehicle routing problems via ensemble with transferrable local policy. IJCAI, 2024.
> [3] Distance-aware attention reshaping for enhancing generalization of neural solvers. IEEE TNNLS, 2025.
> [4] Learning to solving vehicle routing problems via local–global feature fusion transformer. Complex & Intelligent Systems, 2025.
> [5] Instance-conditioned adaptation for large-scale generalization of neural routing solver. arXiv, 2025.
>
> ~~~

---

> ### Author Response · Authors · 2025-11-28
> **Response to Reviewer 55JA - Part 3**
>
> ## Response to Weakness 4:
>
> We will revise our claim to be more precise. Our statement refers specifically to three simultaneous criteria: (i) handling multiple sequence-dependent RWVRP variants (VRPTW, EVRPCS, VRPRS, AVRP) where feasibility depends on complete path history, (ii) scaling effectively to 1000+ nodes, and (iii) within a single architecture without massive variant-specific modifications. Each of the prior studies addresses only subsets of these criteria, for example, RouteFinder (Berto et al., 2025) handles multiple variants but only demonstrates results up to 100 nodes with 11.3-36.7% gaps at larger scales (Table 1); EVRPRL and EVGAT are specialized for electric vehicle routing but achieve 17.8-29.9% gaps on 1000-node instances; POMO and MTPOMO are unified architectures but fail catastrophically at scale (27.6-132% gaps on 1000 nodes, Table 1-2); divide-and-conquer methods (UDC, GLOP) scale well but fundamentally cannot handle sequence-dependent constraints due to the circular dependency between clustering and state computation. So, no prior work simultaneously achieves all three. To clarify, we will revise the claim to: "To the best of our knowledge, SEAFormer is the first learning-based approach to effectively solve 1000+ node instances across multiple RWVRP variants with sequence-dependent constraints (VRPTW, EVRPCS, VRPRS, AVRP) within a single architecture, achieving competitive or superior performance consistently". This more accurately reflects our contribution while acknowledging the substantial prior work in this area.
>
> ## Response to Weakness 5:
>
> While we provide complexity analysis (Equation 6) and empirical validation through ablation studies (Table 3, Table 8, Figure 4), a thorough quantitative analysis on the trade-off between complexity and performance would further strengthen our paper. As summary of the empirical benefit shown in the originally submitted paper, Table 3 demonstrates that CPA outperforms generic sparse attention (Reformer-LSH by 0.3-1.27%, K-means by 3.9%), suggesting polar clustering provides advantages beyond complexity reduction; Figure 3 of the originally submitted paper indicates CPA alone converges slower than POMO, but CPA+Edge-Aware Module (EAM) together converge faster. Table 8 reports the performance of SEAFormer under different CPA hyperparameter settings, while Figure 4 illustrates the performance gap and training memory consumption across these configurations.
>
> For the quantitative analysis on the trade-off between complexity and performance, in the revision, we will conduct comprehensive ablation experiments, isolating each component's contribution and complexity by testing multiple configurations: (i) SEAFormer-NoCPA (standard $O(n^2)$ attention + EAM) to measure EAM's standalone value, (ii) SEAFormer-NoEAM (CPA only, no edge embeddings) to measure polar clustering's standalone value, (iii) SEAFormer-GenericSparse (Reformer + EAM) to test whether any sparse attention mechanism suffices or if polar geometry is essential. Moreover, we will experiment on the number of partitioning rounds R $\in$ {1,2,4}, measuring solution quality, convergence speed, memory consumption, and both training and inference time to quantify the trade-off between complexity and performance. This study will provide a thorough understanding of how CPA impacts scalability, efficiency, and solution quality. As we are still running the experiments, the results will be incorporated into the revised paper as follows: (i) the results for SEAFormer-NoEAM will be added to Figure 4, and (ii) a new subsection (F.5) will be included to present the complexity analysis for each of the settings described above.
>
> ## Response to Weakness 6:
>
> We appreciate the opportunity to clarify this important architectural detail. The edge embedding module does not operate only on depot–customer edges; it embeds both the direction and distance of each edge between every node and its K nearest neighbours. This allows it to represent both depot–customer and customer–customer relationships effectively, where edge embedding from i and j is not identical to that of j to i due to direction (in all problems) or distance (in AVRP). (Appendix J, Page 25 of the originally submitted paper). In Appendix F.4 of the revised paper, we will conduct a sensitivity analysis on K $\in$ {20, 50,100, 200} and report how varying K affects performance, convergence, and both training and inference time, thereby quantifying the trade-off between edge coverage, performance, and computational cost.

---

> ### Author Response · Authors · 2025-11-28
> **Response to Reviewer 55JA - Part 4**
>
> ## Response to Weakness 7:
>
> The limited number of baselines, including the absence of HGS, which has no implementation for EVRPCS or VRPRS, is not an oversight but reflects a fundamental incompatibility between sequence-dependent RWVRPs and methods designed for classical, capacity-constrained VRPs. As discussed earlier, EVRPCS and VRPRS impose state-dependent feasibility conditions such as battery level and remaining travel time, which are absent from standard VRP formulations. As a result, most existing neural or heuristic VRP solvers cannot be directly applied to EVRP-type problems, and many do not support EVRPCS or VRPRS at all.
>
> To ensure a fair comparison, in the revised paper, we will apply a consistent procedure to generate valid EVRPCS solutions from existing VRP methods. Specifically, we first run UDC and HGS as standard VRP solvers, ignoring battery constraints. We then identify any route where a vehicle would run out of battery and apply a simple repair mechanism (UDC-R / HGS-R). For each vehicle in the UDC or HGS solution, and at each customer, we check whether it can reach the next customer and subsequently a charging station. If both are feasible, we follow the planned route; otherwise, we redirect the vehicle to the nearest charging station before continuing. Please note that, since HGS does not natively support EVRPCS, we apply this repair strategy to make it compatible with the EVRPCS setting.
>
> | Method | Size | Infeasibility Rate | Objective | Gap vs. SEAFormer |
> |:-------|-----:|:-------------------|----------:|:------------------|
> | HGS | 100 | 27% | — | N/A |
> | HGS-R | 100 | 0% (repaired) | 16.38 | +0.03% |
> | HGS | 1000 | 100% | — | N/A |
> | HGS-R | 1000 | 0% (repaired) | 51.8 | +13% |
> | UDC | 1000 | 100% | — | N/A |
> | UDC-R | 1000 | 0% (repaired) | 50.9 | +11% |
>
> As shown in our results, VRP solvers with repair strategies yield sub-optimal solutions because they rely on greedy local insertion of charging stops, which often delays charging until it is too late and forces vehicles to take long detours for recharging before they can resume deliveries. An optimal charging decision, however, requires global awareness of the entire route, not just local context. Since VRP solvers optimize primarily for distance, they cannot account for battery-dependent feasibility when choosing the next node. When charging stations are inserted after route construction, the spatial coherence of the route is disrupted. For example, a VRP route A → B → C may become A → B → charging → C, while the optimal EVRPCS solution would be A → charging → B → C.
>
> We will add Appendix G to the revised paper explaining why standard VRP methods fail on RWVRPs, including the experimental results with HGS-R and UDC-R with repairs. We will also include a paragraph explicitly stating that our comparisons are limited to methods that natively support RWVRP constraints, as repairs reduce performance. Additionally, we will add a footnote in Table 1 noting that methods that do not directly account for charging stations or replenishment stops are excluded, since repairing their routes leads to sub-optimal results. For example, HGS-R achieves an objective of 51.8 compared to SEAFormer’s 45.8, a 13% degradation (refer to Appendix G of the revised paper).
>
> In addition, we have conducted a new set of experiments using a real-world EVRPCS dataset presented by [1] and compare SEAFormer against the best-known solutions to demonstrate its effectiveness and quantify the gap to the true optimal solution (refer to our response to Weakness 9). These results will be added to Appendix D of the revised paper.
>
>
> ## Response to Weakness 8:
>
> We appreciate this feedback. While we have provided a complete model parameters in Appendix J of originally submitted paper, training memory consumption in Appendix F.1 of originally submitted paper, and inference times throughout our experimental tables (Tables 1-2 show times ranging from 5 seconds for 100-node instances to 1 minute for 1000-node greedy decoding for a dataset of 100 instances), we agree that a systematic comparison of model parameters and training costs is needed. We will add Appendix L into the revised paper to have a dedicated comparison table in the that includes: (i) model parameters for SEAFormer versus baselines, (ii) training time per epoch and total training cost, and (iii) per-instance inference time.
>
>
> ~~~
> [1] M. Mavrovouniotis, C. Menelaou, S. Timotheou, Christos Panayiotou, G. Ellinas, M. Polycarpou. Benchmark Set for the IEEE WCCI-2020 Competition on Evolutionary Computation for the Electric Vehicle Routing Problem, Technical Report 2020, KIOS CoE, University of Cyprus, Cyprus, March 2020.
> ~~~

---

> ### Author Response · Authors · 2025-11-28
> **Response to Reviewer 55JA - Part 5**
>
> ## Response to Weakness 9:
>
> Thank you for your insightful comment. We have conducted a new set of experiments on real-world datasets for EVRPCS from [1] and VRPTW from [2] for the revision. The EVRPCS dataset contains 24 instances with customer sizes ranging from 29 to 1000 and 4–13 charging stations. We will focus on the 16 instances, each with more than 100 customers and provide a complete set of experiments using SEAFormer and the relevant baselines. The VRPTW dataset from [2] includes 60 instances with 1000 customers. We will update the paper with the results of SEAFormer and relevant baselines on these datasets.  The results of SEAFormer and baselines on these datasets are as below:
>
> ### Real-world EVRPCS Results
> | Method | Gap (%) |
> |:-------|--------:|
> | SEAFormer | 8.2% |
> | SEAFormer-SGBS | 6.5% |
> | EVGAT | 24.5% |
> | EVRPRL | 22.6% |
>
> ### Real-world VRPTW Results
>
> | Method | Gap (%) |
> |:-------|--------:|
> | SEAFormer-SGBS | 7.4% |
> | SEAFormer | 9.1% |
> | DAR | 21.4% |
> | POMO | 47.3% |
> | MTPOMO | 56.1% |
>
> ~~~
> [1] M. Mavrovouniotis, C. Menelaou, S. Timotheou, Christos Panayiotou, G. Ellinas, M. Polycarpou. Benchmark Set for the IEEE WCCI-2020 Competition on Evolutionary Computation for the Electric Vehicle Routing Problem, Technical Report 2020, KIOS CoE, University of Cyprus, Cyprus, March 2020.
>
> [2] Gehring &  Homberger's extended VRPTW benchmark, site: https://www.sintef.no/projectweb/top/vrptw/homberger-benchmark/
> ~~~

---

### Official Review · Reviewer_T5Pk · 2025-10-31

**Soundness:** 2
**Presentation:** 3
**Contribution:** 2
**Rating:** 4
**Confidence:** 4

**Summary:**

This paper presents a Transformer-like framework, called SEAFormer, for solving real-world vehicle routing problems (RWVRP), which integrates both node-level and edge-level information for decision-making. It includes two components, 1) clustered
proximity A=attention (CPA)  for computational efficiency that uses locality-aware clustering and achieving attention with linear complexity; 2) a lightweight edge-aware module that incorporates pairwise relational information into routing decisions for edge-specific constraints. Experimental results show the effectiveness of the proposed method.

**Strengths:**

1. This paper proposes a novel transformer-based framework for solving four real-world vehicle routing variants with diverse problem sizes.
2. The proposed CPA computes attention scores using locality-aware clustering and achieving O(n) complexity while preserving global perspective.
3. Experimental results show the superiority of the proposed method against baselines across various problem sizes and problem constraints.

**Weaknesses:**

1. While CPA and edge modules are well-motivated, the ideas of developing attention with linear complexity and exploiting edge-level information are not novel [1][2][3], and they also build on existing sparse attention and residual fusion ideas (e.g., Reformer, FlashAttention, GAT-based methods).
2. The proposed CPA incorporates some hyper-parameters, e.g., Cluster Size (M) and Partitioning Rounds (R). However, it lacks a systematic analysis of their sensitivity and effects to the model performance.
3. For experimental results, the major superiority of the proposed method seems to be from SGBS inference strategy. For standard VRPs that do not incorporates edge-specific attributes, the superiority of the proposed SEAFormer is not obvious.

[1] Luo F, Lin X, Wu Y, et al. Boosting neural combinatorial optimization for large-scale vehicle routing problems[C]//The Thirteenth International Conference on Learning Representations. 2025.

[2] Meng D, Cao Z, Wu Y, et al. EFormer: An Effective Edge-based Transformer for Vehicle Routing Problems[J]. IJCAI, 2025.

[3] Meng D, Cao, Z, Gao J, Wu Y, et al. UniteFormer: Unifying Node and Edge Modalities in Transformers for Vehicle Routing Problems, NIPS, 2025.

**Questions:**

Please refer to the weakness.

---

> ### Author Response · Authors · 2025-11-26
> **Response to Reviewer T5Pk - Part 1**
>
> Thank you for the valuable feedback. To address your comments, we will include additional experiments that more clearly distinguish SEAFormer’s problem-specific polar clustering and lightweight edge-aware design from generic sparse attention approaches (e.g., Reformer) and GCN-based architectures (EFormer, UniteFormer). We will also expand our comparisons with these recent methods in the experiments to better highlight the performance and scalability differences (addressing Weakness 1). In addition, we will include a complementary ablation study to our existing ablations and sensitivity analyses to further validate the contribution of each component and to examine the sensitivity of each key hyperparameter in greater detail (addressing Weakness 2). We also demonstrate that SEAFormer performs strongly even without the SGBS search method, and we will provide experiments showing that the geometric-aware sparse attention in CPA and the edge-aware module together give the model a clear advantage over the baselines (addressing Weakness 3). We believe these additions will substantially strengthen the clarity and our claims.
>
> ###
>
> ## Response to Weakness 1:
>
> While we acknowledge that we are inspired by prior work on attention mechanisms (studied in Sec 2.3) and edge-based modules, SEAFormer introduces fundamentally different approaches, designed specifically to address routing problems. We are not claiming to invent sparse attention or edge modeling in general, but rather demonstrating how to effectively combine problem-specific spatial sparsity with edge-level embeddings to address real-world routing constraints at scale, where prior methods fail (as shown in Tables 1-2).
>
> **Our proposed Clustered Proximity Attention (CPA) vs. Prior Work:**
>
> Unlike generic sparse attention mechanisms (Reformer's LSH, FlashAttention's memory optimization, Longformer's sliding windows), CPA exploits the geometric structure inherent to routing problems. As described in Section 3 of the paper, our polar coordinate-based clustering reflects how optimal routes naturally form around depot-centric patterns, a domain insight absent from prior methods. Moreover, CPA's multiple partitioning rounds with varying coefficients (Eq. 2 and 3) create complementary spatial perspectives. This differs fundamentally from:
> * Reformer: Uses stochastic LSH that can miss spatial relationships
> * BLEHD[1]: Employs a cross-attention mechanism that can also miss spatial relationships
> * Generic sparse attention: Designed for sequential text, not radial spatial problems
>
> In the ablation study presented in Section 5.3 of the originally submitted paper, we replaced CPA with Reformer, k-means, and grid-based clustering, while keeping the rest of SEAFormer unchanged. Each variant is retrained from scratch using the same training configurations. Table 3 shows CPA outperforms Reformer-LSH (0.3-1.27% gap) and dramatically outperforms K-means (28.7% gap on VRP1000), demonstrating our approach's superiority for routing tasks.
>
> **Our Edge-Aware Module vs. Prior Work:**
>
> EFormer [2] and UniteFormer [3] are recent VRP methods that combine Graph Convolutional Networks (GCNs) for edge embeddings with attention-based node representations, and rely on parallel decoding to generate high-quality solutions. In contrast, SEAFormer adopts a much more lightweight design: it does not use a dedicated GCN network for edge embeddings and step-by-step attention-based decoding based on edge embeddings as it is done by Eformer and UniteFormer. Instead, it employs a simple residual module to integrate node embeddings with edge features, and uses an MLP to produce an edge-aware heatmap, only once in the beginning decoding process, resulting in a lighter model with a higher performance. This design offers several practical advantages:
> * Faster convergence (Figure 4 of the revised paper)
> * Smaller model with fewer parameters with respect to Eformer and UniteFormer (will be reported in the revised paper)
> * Lower memory consumption (Figure 6 of the revised paper), enabling training on significantly larger problem sizes, and improved solution quality at identical inference speed.
>
> Our initial investigations for the revised paper on VRP100 show that SEAFormer achieves a cost of 15.72, outperforming both UniteFormer (15.74) and EFormer (15.77). We will include reference [1-3] and the differences in Appendix 'additional related work’. Also, we will expand the paper by reporting comprehensive results for EFormer and UniteFormer on additional problem sizes with 500 and 1000 nodes, and include these results in Table 2. Furthermore, as we are still running the experiments, we will include an additional ablation experiment in Figure 4 of the revised Section 5.3, where the edge-aware module is paired with a full-attention mechanism to more clearly isolate and quantify its contribution. We will also notify you of the new results.

---

> ### Author Response · Authors · 2025-11-26
> **Response to Reviewer T5Pk - Part 2**
>
> ## Response to Weakness2:
>
> We provided extensive ablation studies in Appendix F (pp. 20–22) of the originally submitted manuscript, including the results for varying M and R, covering performance, memory consumption, and training convergence. Below, we are providing the details of their sensitivity and effects on the performance discussed in the originally submitted paper. In addition, a new set of experiments examining the impact of these parameters on both inference and training time will be included in the revised manuscript to more clearly characterize the sensitivity of the key hyperparameters and their effects. As we are still running the experiments, the results will be added in Appendix F, and we will notify you of the new results.
>
> The sensitivity and effects for varying M are presented in Table 8, which shows that increasing the number of partitioning rounds (R) from 1 to 4 improves the optimality gap from 1.19% to 0.56% when the cluster size (M) is fixed at 50. Using four partitioning rounds with M=50 also yields 2× faster convergence compared with POMO: SEAFormer reaches POMO’s 1,000-epoch performance in only 500 epochs, under identical settings and hardware.
>
> Figure 4a further shows that performance saturates around R=2, with diminishing returns beyond this point, while Figure 4b quantifies the memory–quality trade-off: each additional round increases memory usage by approximately 5–15%, yet provides about 0.2% improvement in solution quality.
> Cluster size (M) is also examined in Appendix F. Figure 4b shows that increasing M from 20 to 200 raises training memory by only ~10%, while Table 8 demonstrates that increasing M from 20 to 50 improves solution quality by ~1% relative to the best achievable objective.
>
> Taken together, these results indicate that R=2 (with boundary smoothing) and M=50 provide the best balance of accuracy and memory efficiency for SEAFormer.
>
> ## Response to Weakness3:
>
> SEAFormer without SGBS demonstrates strong performance, specifically for solving the different variants of the VRP. The proposed seaformer architecture is a unified approach to address a comprehensive range of real-world important VRP routing constraints (not just the classical VRP) at scales within a single architecture. Therefore, the performance of SEAFormer (both with and without SGBS) is as good as all the recent VRP solutions, and SEAFormer is the single solution that can answer all the VRP variants mentioned in the paper and outperforms all the variant-specific existing approaches. The inclusion of SGBS simply further boosts the already existing performance of SEAFormer, especially for larger instances. For example, on VRP1000, SEAFormer without SGBS already surpasses strong recent methods such as LEHD (0.62% vs. 1.05% gap to HGS),  RELD (0.62% vs. 2.36% gap to HGS), and L2C (0.62% vs. 0.82% gap to HGS). Moreover, SEAFormer achieves superior results on VRP5000 and VRP7000 even without SGBS, outperforming UDC by 0.65% and 0.26%, respectively. On VRPTW, SEAFormer significantly outperforms all baselines, achieving at least a 4% improvement over the time-consuming OR-Tools solver.
>
> This strong performance is because CPA offers geometric-aware sparse attention, and the edge-aware module captures pairwise relational patterns that node embeddings alone cannot represent. While CPA alone improves scalability, it converges more slowly. Incorporating the edge module consistently accelerates convergence and improves training stability, making the CPA + edge-aware module combination a complementary and effective design for RWVRPs. Moreover, the edge-aware module enhances robustness. For instance, Appendix E shows that SEAFormer generalizes effectively across different distributions without retraining, whereas models lacking edge information degrade.
>
> Please note that the edge-aware module is applied to all problem instances, not only to those with explicit edge-specific attributes such as AVRP. In our formulation, each edge is described using two components: the Euclidean distance and the cosine similarity between node coordinates, which encodes the directional orientation of the edge. When combined, these features provide richer spatial information: the Euclidean distance captures “how far” the vehicle must travel, while the cosine similarity captures “how much the direction changes.” This pairing strengthens the model’s ability to reason over spatial structure, even in settings where edge attributes are not explicitly defined.
>
> We will include an ablation study over SEAFormer without hiring an edge-aware module to demonstrate that edge embeddings play a crucial role in its superior performance. These results will be added to Figure 4 of the revised paper and Appendix F.4.

---

### Author Response · Authors · 2025-11-14
**Author Acknowledgment of Reviews**

Dear Reviewers and AC,

Thank you for your time and the thoughtful, constructive feedback. We sincerely appreciate the time and attention you have dedicated to evaluating our work. We are currently conducting additional experiments and preparing a detailed revision plan addressing the points raised. A comprehensive response will be provided within the next few days.

We look forward to constructive discussions in the coming weeks.


Best regards,

Authors of paper 15891

---

### Author Response · Authors · 2025-12-04
**Brief summary of our response to reviewers**

Dear Area Chair,

We sincerely thank the reviewers for their thoughtful and detailed evaluations, and we appreciate your effort in reassessing our work. We have addressed every feedback through substantial clarifications, improved presentation, and extensive new experimental results and analyses. A concise overview of the reviewers’ main feedback and our corresponding responses is as follow:

**Feedback 1:** Insufficient clarity on the innovation over existing sparse/local attention methods.

**Response:** We clarify that CPA introduces a polar coordinate-based clustering mechanism in the encoder, embedding a pattern-driven inductive bias that mirrors how near-optimal routes naturally form around the depot. This design is fundamentally different from existing sparse attention (e.g., Reformer’s LSH) and decoder-level distance-penalty methods (e.g., DAR). Table 1 and Table 2 has been updated with new results.

**Feedback 2:** Performance gains may mainly stem from SGBS; unclear gain from designed components.

**Response:** We show that SEAFormer achieves strong results without SGBS (e.g., 0.62% optimality gap on VRP1000, outperforming strong baselines such as L2C and RELD). New ablation studies also demonstrate that CPA and the edge-aware module are complementary, improving stability and solution quality even on classical VRPs. Figure 4, Appendix F.4, and Appendix F.5 shows results of new studies.

**Feedback 3:** Missing comparisons with recent strong baselines (EFormer, UniteFormer, DAR) and classical solvers (HGS, UDC).

**Response:** We now include full comparisons against EFormer, UniteFormer, and DAR on VRP and VRPTW across 100-1000 nodes (Tables 1 and 2), showing consistent improvements. We also clarify in Appendix G why results of VRP specific methods (such as HGS and UDC) are not directly applicable to state-dependent RWVRPs.

**Feedback 4:** Lack of real-world experiments.

**Response:** We add results on two real-world datasets (VRPTW and EVRPCS, in addition to VRP), showing that SEAFormer achieves superior performance, including improvements over DAR on real-world VRPTW instances. These results are shown in Appendix D.3 and D.4 of the revised paper.

**Feedback 5:** Missing comparisons on model size and training time.

**Response:** We now provide detailed parameter counts and training-time comparisons in the revised paper, Appendix L.

**Feedback 6:** Insufficient analysis of CPA hyperparameters (M and R).

**Response:** We expand the sensitivity study of M and R in the appendix. The results quantify performance-memory trade-offs and show that R = 2 and M = 50 offer the best balance. Appendix F.4 and F.5 demonstrate the results.

We believe the revised manuscript, with clearer definition of our contributions and the inclusion of comprehensive new experiments, fully addresses all reviewer feedbacks.

Best regards,

Authors of paper 15891

---

### Meta-Review · Area_Chair_reEA · 2026-01-07

**Summary:**

This paper proposes SEAFormer, a neural solver for real-world vehicle routing problems (RWVRPs). The key ideas are (1) Clustered Proximity Attention (CPA), which uses locality-aware (polar-coordinate) clustering to reduce attention cost and make solving large instances feasible, and (2) a lightweight edge-aware module to inject edge information such as asymmetric costs. Overall, model is able to scale to 1000+ nodes with sequence-dependent constraints and reported results are strong on multiple VRP variants.

**Reviewer Concerns:**

During the rebuttal, the authors addressed most concerns. They tried to clarify the intended novelty: CPA is positioned as an encoder-side, routing-specific geometric inductive bias, different from generic sparse attention and decoder-side distance-penalty attention. They also argued that the gains are not mainly from SGBS, by highlighting competitive greedy-decoding results. The authors further strengthened the empirical side by adding (or committing to add): (1) broader baseline comparisons, (2) efficiency reporting, and (3) real-world dataset results, which support the “real-world” claim better than before.
However, some concerns still remain: (1) novelty: the concept of "linear attention + edge info” for neural routing solvers is not new; (2) generality: reviewers questioned extension beyond single-depot Euclidean settings. The authors provided reasonable directions for distance-matrix inputs and MDVRP extensions, but these are mostly conceptual and not implemented yet; (3) the claimed linear complexity depends on treating R and M as constants, and reviewers asked for a clearer discussion of boundary cases.

**Reviewer Scores:**

2/4 reviewers would maintain their positive evaluation, while it is not clear if the other two reviewers would be willing to change their evaluation, since their concerns regarding novelty have not been fully resolved.

---

### Decision · Program_Chairs · 2026-01-26

Reject